# Genetic variability in microbial eukaryotes reshapes marine biodiversity assessment in the age of amplicon sequencing

Sarah Trubovitz[1,2]*, Miguel M. Sandin[3], David A. Caron[1]

**1** Department of Biological Sciences, University of Southern California, Los Angeles, California, United States of America, **2** Ocean Sciences Department, University of California-Santa Cruz, Santa Cruz, California, United States of America, **3** Institut de Biologia Evolutiva (CSIC-Universitat Pompeu Fabra), Barcelona, Spain

* strubovi@ucsc.edu

## Abstract

Rapidly improving DNA sequencing technology has revolutionized our ability to efficiently survey the biodiversity of microbial life. We are now equipped to investigate protistan richness and community dynamics on scales that would not have been imaginable with traditional observational methods. However, for most taxa the relationship between DNA sequences and morphologically-defined species is poorly understood, and morphology has remained the cornerstone of taxonomy for centuries. To better utilize the wealth of sequence data being collected, we must understand how it relates to entities such as individuals, populations, and species. Here we use a combined microscopy and sequencing approach to unveil the striking intragenomic and intraspecies genetic variation in one group of ecologically-important marine protists, the polycystine Radiolaria. Long-read 18S rRNA gene amplicon data from 173 isolated and morphologically-identified radiolarians showed that the vast majority (90%) yielded multiple sequence variants per specimen. Furthermore, every morphospecies analyzed displayed a range of different genetic signatures. Intraspecies genetic variability was expressed as specimens having different assemblages of ASVs, different dominant ASVs, or having no ASVs in common with other specimens of the same morphospecies. By integrating morphological and molecular information, we begin to parse the genetic richness of Radiolaria in ocean environments, as well as illuminate relationships between taxa, and their poorly-known life stages. Our findings emphasize the need to account for protists' taxon-specific sequence variability, particularly their intragenomic and intraspecies genetic variation, in interpreting metabarcoding diversity survey data.

**Data availability statement:** Sequence data underlying the analyses in this article are available from Zenodo (DOI: 10.5281/zenodo.15550382). The raw, unprocessed sequence data are also available for download from the NCBI Sequence Read Archive (SRA) with sample accession numbers SAMN48798827 – SAMN48798999 (under BioProject PRJNA1269742).

**Funding:** This project was supported by the National Science Foundation Postdoctoral Research Fellowship in Biology Award #2109767 (to ST) and the Simons Collaboration on Ocean Processes and Ecology p49802 (to DAC). MMS was supported by a postdoctoral fellowship from the Beatriu de Pinós programme of the Government of Catalonia's Secretariat for Universities and Research of the Ministry of Economy and Knowledge (grant #2021BP00068).

**Competing interests:** The authors have declared that no competing interests exist.

## Introduction

DNA sequencing has provided a wealth of information that has been transforming our understanding of marine microbial eukaryote species richness, biogeography, community composition, and evolutionary relationships [1,2]. In the field of ecology, high-throughput sequencing has enabled the detection of microorganisms that are otherwise too rare, fragile, fastidious, or morphologically indistinct to be readily observed with classical approaches of microscopy and culture. However, at this new frontier of DNA sequencing comes new challenges regarding data interpretation and integration with previous knowledge [3]. Metabarcoding is a highly-efficient technique for measuring biodiversity in environmental samples, which involves amplifying and sequencing a specific region within a chosen marker gene that is common to all targeted taxa (typically the 18S rRNA gene for microbial eukaryotes, a.k.a. "protists"). In this way, metabarcoding reveals the genetic richness of a sample, but ultimately the taxonomic assignment and interpretation of those gene sequences is limited by the quality of curated sequence databases (such as the Protist Ribosomal Reference Database, PR2 [4]). Due to the laborious nature of linking morphological and molecular data, these databases so far encompass only a small fraction of protistan richness, resulting in many metabarcoding sequences with dubious taxonomic identities and little consensus as to what magnitude of sequence variation should delineate meaningful biological units. Furthermore, while morphology-based taxonomy remains the gold standard in protistology, it is known to be imperfect, especially for rare, poorly-studied, and morphologically-indistinct taxa [3]. This poses the question: how can we best utilize genetic sequence information that is often only loosely tethered (if at all) to the morphological concepts that have historically been considered the common denominator in biological sciences?

Discrepancies between morphology-based and sequence-based taxonomic assessments for some protistan groups can be confounding, and for one ubiquitous group of marine planktonic protists, the polycystine Radiolaria [5], these differences are particularly stark. More than a century of taxonomic inventorying and description of polycystine radiolarian specimens in marine sediments, plankton nets, and sediment traps has yielded approximately 445 formal morphological species [6] and perhaps up to 600–800, counting those that await formal description: most of these in orders Nassellaria and Spumellaria [5]. By contrast, the Tara Oceans expedition revealed an astounding ~6,600 polycystine genetic operational taxonomic units (OTUs) during a sampling expedition that targeted the sunlit ocean alone, and the vast majority of those OTUs were assigned to order Collodaria (another important polycystine group) rather than Nassellaria or Spumellaria [7]. This underscores how protistan morphological species descriptions cannot necessarily be expected to align with other modes of taxonomic delineation, including concepts based on genetic similarity [8,9,10]. There is thus an urgent need to probe this mismatch between morphological and molecular richness so we can improve interpretations of the sequence datasets in genetic diversity surveys.

Beyond epitomizing the disconnect between protistan morphological and molecular diversity, polycystine radiolarians are an ideal study system because their

relatively large size (~20–250 µm for solitary taxa and typically mm–cm for colonial taxa) permits simultaneous morphological and molecular analyses, and their diverse array of siliceous skeletal structures facilitates high-resolution morphological groupings. Additionally, radiolarians play integral roles in marine food webs and biogeochemical cycles, which necessitates a better understanding of their biodiversity. They are abundant in ocean water columns from the surface to abyss, from the tropics to the poles [11], occupying many ecological niches as mixotrophs, herbivores, predators, and detritivores [5,12]. They have also been shown to contribute significantly to the biological ocean carbon pump [13,14,15,16,17,18,19] and the silicon cycle [19,20]. Despite their importance, only a small fraction of polycystine radiolarian morphospecies have been linked to their genetic sequence information [21,22,23,24,25,26,27]. In addition, very few studies have explored radiolarians' intragenomic variation [23,28] (which in this manuscript refers to gene sequence variation within the genome of an individual specimen), or potential cryptic species [29,30] (which refers to genetically distinct subgroups within a morphologically-defined species). Here we build on emerging genetic information for polycystine Radiolaria by combining morphological and molecular analysis of 173 isolated specimens collected from the North Pacific Subtropical Gyre. We employ long-read 18S rDNA amplicon data to investigate the genetic variability within individual specimens and morphospecies, explore phylogenetic relationships among taxa, and discuss the implications of these findings for deciphering protistan diversity datasets in the age of amplicon sequencing.

## Results and discussion

### The dataset

Our dataset included 18S rDNA long-read amplicon sequence variants (ASVs) from 173 polycystine radiolarian specimens belonging to 30 morphologically classified taxa. DNA from each individual was extracted, amplified, and underwent circular-consensus sequencing with Pacific Biosciences technology while morphology-based taxonomy was assigned via microscopy. After bioinformatic processing, a total of 1,224 unique ASVs met criteria for analysis: a minimum of three reads in at least one specimen and taxonomically assigned to the same family as the morphologically-identified specimen they came from (see "Materials and methods"). The specimens in our dataset represented all three polycystine radiolarian orders: Collodaria (95 specimens), Nassellaria (36 specimens), and Spumellaria (42 specimens). Out of these, the Collodaria displayed the highest genetic richness (836 ASVs) despite including the lowest number of morphotaxa (9). Conversely, Nassellaria specimens had the lowest genetic richness (187 ASVs) but highest number of morphotaxa (11). Spumellaria specimens were comprised of 201 total ASVs from 10 morphotaxa. Taxonomic names of all specimens are listed in "Materials and methods," photographic documentation is provided in S1 Fig, and ASV occurrence charts in S1 Data.

### Individual radiolarians contained multiple ASVs

After stringent taxonomy and abundance-based vetting of ASV results (see "Materials and methods"), we found that 156 out of 173 radiolarian specimens (90%) exhibited multiple ASVs, with a median number of 12 ASVs per specimen (Fig 1a). This indicates that intragenomic variation was pervasive among the specimens we analyzed; however, the number of ASVs per specimen varied by taxonomic order. Collodaria specimens showed the greatest genetic richness, with a median of 32 ASVs per specimen and a relatively large range in values (1–141 ASVs per specimen) (Fig 1a). Nassellaria and Spumellaria had fewer ASVs per specimen, with median values of 3 and 5.5, respectively. Nassellarian intragenomic richness ranged from 1–57 ASVs and Spumellaria ranged from 1–40 ASVs per specimen (Fig 1a). The only family that did not yield multiple ASVs per specimen was Lophophaenidae (Nassellaria); *Lophophaena hispida* and *Peromelissa thoracites* each contained a single ASV (S1 Data).

When the hypervariable V4 region was extracted from each long-read ASV (see "Materials and methods"), intragenomic richness of specimens in each taxonomic order was effectively reduced by half. There was a median number of 6 V4-region sequence variants per specimen, with Collodaria showing a range of 1–57 variants (median = 16), Nassellaria

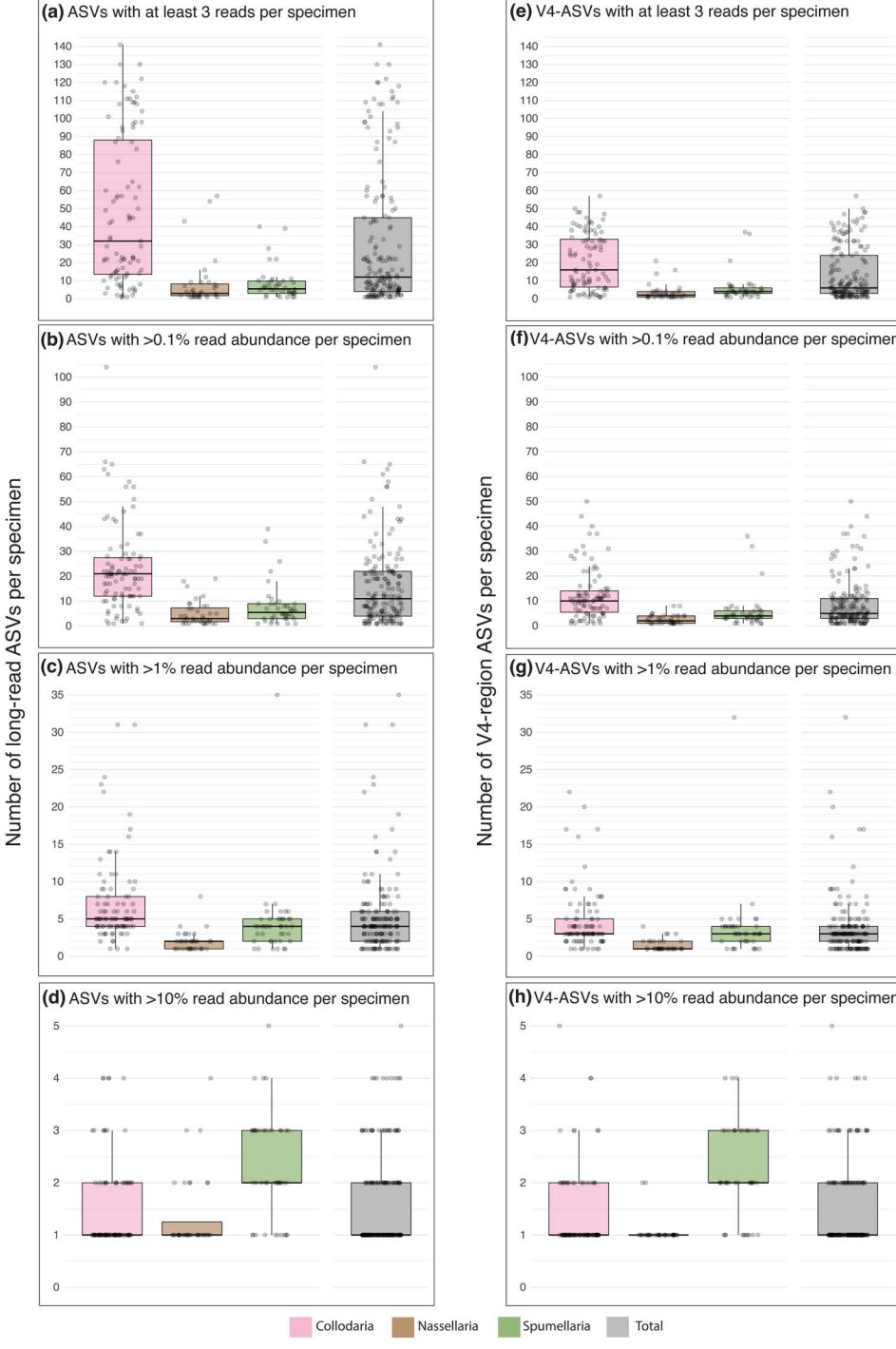

**Fig 1. Number of ASVs per specimen among three orders of polycystine Radiolaria.** Each dot represents one specimen; panels show the distribution of ASVs per specimen in Collodaria (pink), Nassellaria (brown), Spumellaria (green), and all taxa combined (gray). These include: the total number of ASVs that passed all vetting procedures described in "Materials and methods" **(a)**; the subset of ASVs which constituted at least 0.1% of the number of reads per specimen **(b)**; ASVs that made up at least 1% of reads per specimen **(c)**; and ASVs comprising at least 10% read abundance per specimen **(d)**. Panels **(e–h)** display corresponding intragenomic richness information for only the extracted V4 regions of the full-length ASVs, respectively. Boxplots denote the median, interquartile range, maximum, and minimum values (outliers excluded). All data underlying this figure is provided in S1 Data.

ranged from 1–21 (median = 2), and Spumellaria ranged from 1–37 sequence variants per specimen (median = 4) (Fig 1e). Therefore, while short-length V4 ASVs masked a significant portion of the genetic richness captured by full-length ASVs, intragenomic variation was still evident in each polycystine order albeit to a lesser degree. This result suggests that metabarcoding surveys utilizing relatively short-reads of hypervariable regions such as the V4 or V9 (e.g., [7]) are not able to detect the same level of genetic richness that can be revealed with long-read sequence data.

The biological significance of protistan ASVs and their relationship to concepts such as specimens, populations, or species, remains a complex open question [3]. Previous work has suggested that ASVs almost certainly capture some intraspecies variation [3,10], but our results expand those findings by demonstrating that radiolarian ASVs can constitute a great deal of intragenomic variation as well. The fact that all three radiolarian orders demonstrated intragenomic variation indicates this phenomenon is not limited to species with a colonial life habit nor the presence of multiple nuclei, traits which frequently occur in Collodaria but have not been observed in any Nassellaria or Spumellaria [5,31]. Rather, it is probable that individual nuclei contain intragenomic variation for all groups included in this study (except evidently the Lophophaenidae, but more research is needed). Furthermore, because the average specimen in our dataset yielded 12 unique ASVs, intragenomic variability is likely a significant source of the total genetic richness being measured in metabarcoding surveys of protistan diversity. The intragenomic variability we observed ranged widely among taxa, even those belonging to the same order. Therefore, while multiple ASVs could indicate multiple individuals (or multiple species) in some taxonomic groups, in others it appears to reflect the intragenomic variation within a single individual. This suggests that accurately interpreting the significance of ASV richness will require taxon-specific knowledge of genomic variability.

Next, we examined ASV read abundances within specimens to assess the prominence of intragenomic variants. Thresholds were applied to determine the number of ASVs contributing at least 0.1%, 1%, or 10% of the total reads within each specimen (Fig 1b–d, respectively). These more abundant subsets of ASVs within each specimen presumably reflect their higher frequencies within the genome, potentially having a greater impact on the organism's function and fitness compared to rarer ASVs [32]. Collodaria specimens yielded a median of 21 ASVs (range: 1–104) that comprised at least 0.1% relative read abundance (Fig 1b). At a minimum of 1% relative read abundance (Fig 1c), this decreased to a median of 5 ASVs per collodarian (range: 1–31), and at 10% (Fig 1d) collodarians exhibited between 1 and 4 ASVs with a median value of 1 per specimen. Nassellaria and Spumellaria had fewer ASVs per specimen than Collodaria at every relative abundance threshold except for 10%, where Nassellaria exhibited the same median and range as Collodaria, and Spumellaria exceeded both groups by 1 ASV in terms of median and range (2 and 5 ASVs, respectively) (Fig 1d).

The partitioning of reads among ASVs was investigated in greater detail by examining the relative read abundance of the top ten ranking ASVs per specimen in each order (Fig 2). Nassellaria specimens, on average, had a single, highly-dominant ASV, while a relatively low number of subdominant ASVs within their genomes comprised a minor percentage of reads. Among Nassellaria, the top ranking ASV made up a median of 92% of total reads within each specimen, versus 77% in Collodaria and only 60% in Spumellaria. Compared to the other orders, Spumellaria showed a relatively equitable distribution of reads among their dominant and top few subdominant ASVs. For example, the 2nd rank ASV in Spumellaria comprised 23% of total reads per specimen on average, whereas 2nd rank ASVs in Collodaria and Nassellaria only made up 5% and 4% of reads per specimen, respectively. Among the extracted V4 region ASVs, the different partitioning patterns between Spumellaria and the other two orders were even more pronounced (S3 Fig). Collodaria and Nassellaria specimens had top ranking V4-ASVs make up 88% and 98% of their total reads (median values), respectively, whereas Spumellaria specimens only had a median of 61% of total reads in their top-ranking V4-ASV. Based on these findings, we speculate that intragenomic variants in Spumellaria may tend to be more functionally important compared to those in other orders, even though Collodaria yielded the highest numbers of intragenomic variants overall.

One possible explanation for the intragenomic variation observed in radiolarians is that they could be polyploid. Past imaging studies have suggested that some polycystines exhibit polyploidy (e.g., [33]); however, we are not aware of any definitive evidence that polyploidy is confirmed or common in polycystine radiolarians, and our dataset is not sufficient to

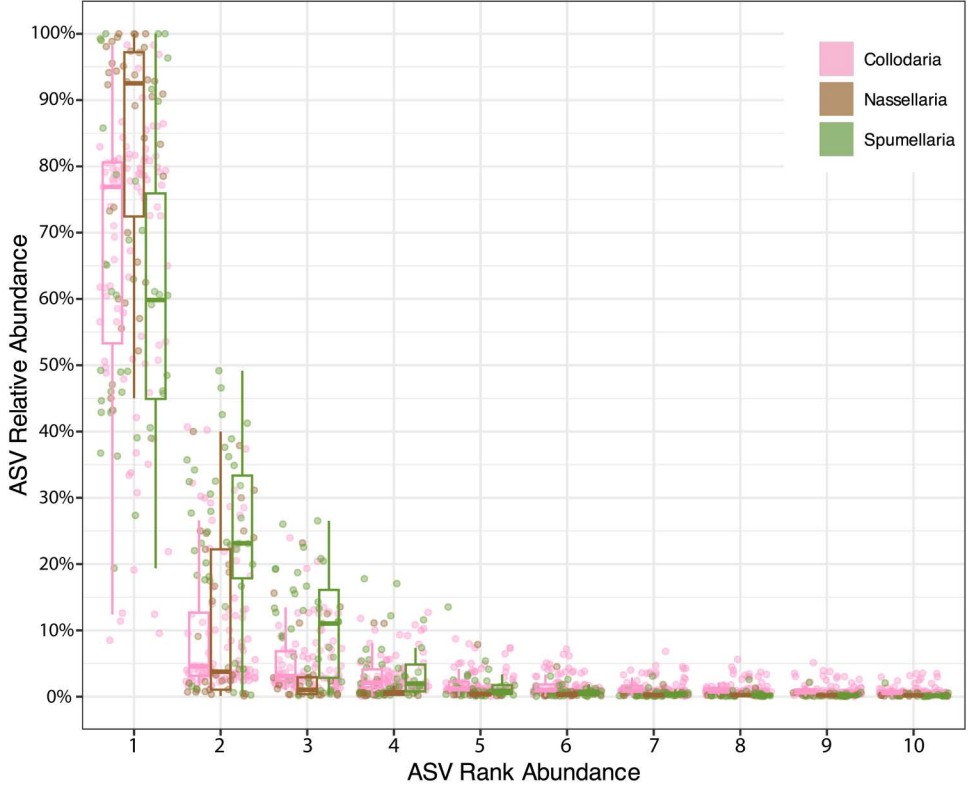

**Fig 2. Distribution of relative read abundances among the top 10 ranking ASVs per specimen.** Dots represent relative ASV abundances (percent of total reads yielded by each specimen) at each designated rank (1st–10th) for all 173 specimens. Boxplots denote the median, interquartile range, maximum, and minimum values (outliers excluded) of ASV relative abundances for each order (Collodaria = pink; Nassellaria = brown; Spumellaria = green). Data underlying this figure is in S1 Data.

test this hypothesis. Polyploidy has been better documented in other members of Rhizaria, including Acantharia, Phaeodaria, and Foraminifera [34,35], as well as other protistan groups including ciliates [36] and diatoms [37].

Another non-exclusive explanation for the intragenomic variation we observed could be the presence of high copy numbers of the 18S rRNA gene. Some polycystines (Collodaria) have been shown to have astoundingly high gene copy numbers per cell (estimated at 5,770 ± 1,960 SSU rDNA copies for solitary specimens and 37,474 ± 17,799 for cells isolated from colonies [38]), similar to groups such as dinoflagellates and ciliates [39,40,41], although it has been suggested that Nassellaria and Spumellaria have fewer copies [28]. Copy number has been shown to positively correlate with cell length across marine protist clades [38] as well as with intragenomic variation in ciliates [42]. Thus, copy number may have played a role in the high intragenomic richness we observed in Collodaria, and may also partly explain the relatively lower intragenomic richness observed in other polycystine groups that tend to have smaller cell sizes.

Regardless of the underlying cause, differences in intragenomic richness and ASV read abundance structure across taxa have major implications for diversity interpretations in metabarcoding studies. High-throughput environmental DNA sequencing yields an immense quantity of data that is broadly representative of diversity in a sample. However, both the richness and relative abundance of ASVs per specimen was not uniform across major taxonomic groups, so their metabarcoding results should not be interpreted uniformly to avoid misleading conclusions regarding diversity. Global scale genetic surveys of marine protistan species richness have revealed such great diversity of radiolarian molecular taxa [7] that they exceed the number of taxa in morphology-based surveys of similar scale by more than an order of

magnitude [43]. The high degree of variability we observed in radiolarians' intragenomic richness and ASV abundance structure may underlie some of the apparent 'species' richness and relative abundances observed in metabarcoding diversity surveys.

The prevalence of this situation may extend to other protistan taxa as well. Previous work has documented intragenomic variation in Nassellaria and Spumellaria [23,28], and in polycystine radiolarians' sister clades, the Acantharia [23] and planktonic Foraminifera [44,45]. There is also growing evidence that distally-related protistan clades exhibit intragenomic variation (e.g., [42,46,47]), which taken together may partly explain the staggering number of sequence variants observed in marine metabarcoding diversity surveys (i.e., [7]). Intragenomic variation has not been explored for the vast majority of taxa nor is it typically accounted for in richness interpretations of metabarcoding data [48]. It is thus imperative that we gain a better understanding of the magnitude and structure of intragenomic variation across groups, so that we can more accurately apply metabarcoding data to address important questions in ecology, including the unified understanding and detection of biological units, like individuals, populations, and species.

## Specimens of the same morphospecies showed varying ASV assemblages

Morphospecies of which we were able to collect multiple individual specimens were analyzed for intraspecies genetic variation. This analysis included a total of 14 morphospecies, and all three polycystine orders were represented. Intraspecies genetic variation (here defined as different assemblages of ASVs within specimens of the same morphospecies) was found in every morphospecies we examined, to varying degrees (summarized in Table 1).

The most thoroughly-sampled morphospecies in this study was the colonial collodarian, *Collosphaera huxleyi* (51 specimens). It yielded a combined total of 562 ASVs, with a median of 87 ASVs per specimen (S1 Data). Many of these ASVs (41%; 232) were shared by at least two specimens, and some ASVs (7%; 38) were shared by more than half of specimens. But, only one ASV was shared by every specimen in the dataset (Fig 3) and this ASV was dominant in 48 of 51 specimens (S2 Fig; S1 Data). Generally, there was a positive correlation between the total read abundance of an ASV and the number of specimens that shared it, but this was not always the case, indicating that some specimens yielded high abundances of ASVs that were absent in other specimens (Fig 3). The fact that hundreds of unique sequence variants could be attributed to one morphospecies is surprising, especially given that *C. huxleyi* exhibits relatively simple and consistent morphological characteristics (S1 Fig) and lacks any known niche differentiation.

All of our specimens were collected during summer 2022 in approximately the same location at ~5m depth (see "Materials and methods"). Therefore, temporal or spatial separation cannot easily explain the remarkable magnitude of gene sequence variation we observed in *C. huxleyi* specimens. The life cycle of *C. huxleyi* is poorly understood, but each specimen we collected was colonial, with 10s–100s of skeletonized cells matching the formal species description (*sensu* Strelkov and Reshetnyak [49]). It was common, however, for colonies to contain cells of varying sizes, sometimes in the process of dividing (also observed by Strelkov and Reshetnyak; Fig 19 in [49]), so we cannot rule out the possibility that differing stages of the colonies or the cells within them could be related to the genetic variation we observed. Alternatively, this intraspecific variation could be due to random neutral mutation that was not related to life stage. Although there was a lack of distinct genetic subgroups in this population (i.e., all specimens shared one, usually dominant, ASV) the fact that some specimens had different dominant ASVs and most ASVs were shared by less than half of specimens, could suggest ongoing genetic drift potentially leading to speciation, including the possible emergence of cryptic species. Either way, the multitude of ASVs observed among the 51 *C. huxleyi* specimens raises questions as to how to interpret that sequence diversity with respect to the morphologically-defined taxon.

The other morphospecies in this study also demonstrated intraspecies genetic variation, but they did so with differing patterns of ASV presence and abundance (Table 1; Figs 4–6). Out of the 14 morphospecies examined, only one (*Pterocanium praetextum*) yielded the same dominant ASV in all specimens, although only 3 specimens could be examined (Fig 5d). There were also two subdominant ASVs found in this species; one was shared by all specimens and the other shared

**Table 1. Categories of intraspecies genetic variation observed in this study.**

| | Morphospecies (# specimens) | The same ASV was dominant in all specimens | Some or all specimens had different dominant ASVs, but at least 1 ASV was shared by all | There was no ASV shared by every specimen of the morphospecies |
|---|---|---|---|---|
| **Collodaria** | *Acrosphaera murrayanna* (6) | | | X |
| | *Collosphaera huxleyi* (51) | | X | |
| | *Disolenia zanguebarica* (2) | | X | |
| | *Procyttarium primordialis?* (3) | | X | |
| | *Siphonosphera socialis* (6) | | | X |
| | *Thalassicolla nucleata* (10) | | | X |
| **Nassellaria** | *Acanthodesmia vinculata* (2) | | | X |
| | *Eucyrtidium hexagonatum* (2) | | | X |
| | *Pterocanium praetextum* (3) | X | | |
| | *Pterocorys zancleus* (19) | | | X |
| **Spumellaria** | *Dictyocoryne profunda* (6) | | | X |
| | *Dictyocoryne truncatum* (3) | | | X |
| | *Didymocyrtis tetrathalamus* (10) | | | X |
| | *Euchitonia elegans-furcata* (5) | | X | |

by two out of three specimens. *P. praetextum* thus showed a relatively low level of intraspecific genetic variability, though specimens did not have identical ASV compositions.

By contrast, other morphospecies showed higher degrees of genetic variation than *P. praetextum*. In *Procyttarium primordialis*? (Fig 4e), some specimens yielded different dominant ASVs (dark blue shades in heatmap), although any ASV that was dominant in one specimen was at least present in the others (light blue shades in heatmap). Even greater genetic variation was observed in *Collosphaera huxleyi* (S2 Fig), *Disolenia zanguebarica* (Fig 4b), and *Euchitonia elegans-furcata* (Fig 6c). For these species, the dominant ASV in some specimens was entirely absent in others (white color in heatmaps).

Most morphospecies in this study included some specimens that were not linked by any shared ASVs at all, appearing to constitute separate genetic subgroups. This was the case for *Acrosphaera murrayanna* (Fig 4a), *Siphonosphaera socialis* (Fig 4c), *Thalassicolla nucleata* (Fig 4e; but see caveats regarding *Thalassicolla nucleata* in subsequent section), *Pterocorys zancleus* (Fig 5a), *Acanthodesmia vinculata* (Fig 5b), *Eucyrtidium hexagonatum* (Fig 5c), *Dictyocoryne profunda* (Fig 6a), *Dictyocoryne truncatum* (Fig 6a), and *Didymocyrtis tetrathalamus* (Fig 6b). These morphospecies each contained at least one representative specimen, or group of specimens, that had an entirely different set of ASVs than other specimens. These cases in which no ASV was shared across all representatives of a morphospecies could be indicative of cryptic species. Notably, the morphological subspecies of *Didymocyrtis tetrathalamus* (*D. t. tetrathalamus* and *D. t. coronatus*; Fig 6b) were not found to be genetically distinct, as they shared some ASVs. Therefore, while *Didymocyrtis tetrathalamus* contained multiple genetic subgroups, the subgroups did not appear to directly correspond to the morphological distinctions defining the two subspecies. Instead, their morphological differences may be attributable to a gene other than 18S or potentially an epigenetic mechanism. Genetic subgroups have previously been reported in another spumellarian morphospecies, *Spongotrochus glacialis*, but in that case genetic differences corresponded with habitat partitioning and a minor difference in skeletal morphology (spine length) [30].

These examples demonstrate that intraspecies genetic variation was the norm rather than the exception among Collodaria, Nassellaria, and Spumellaria. Moreover, a single dominant ASV was rarely shared among all representatives of a given morphospecies (only 1 out of 14 species) and most morphospecies (9 out of 14) included some specimens with no

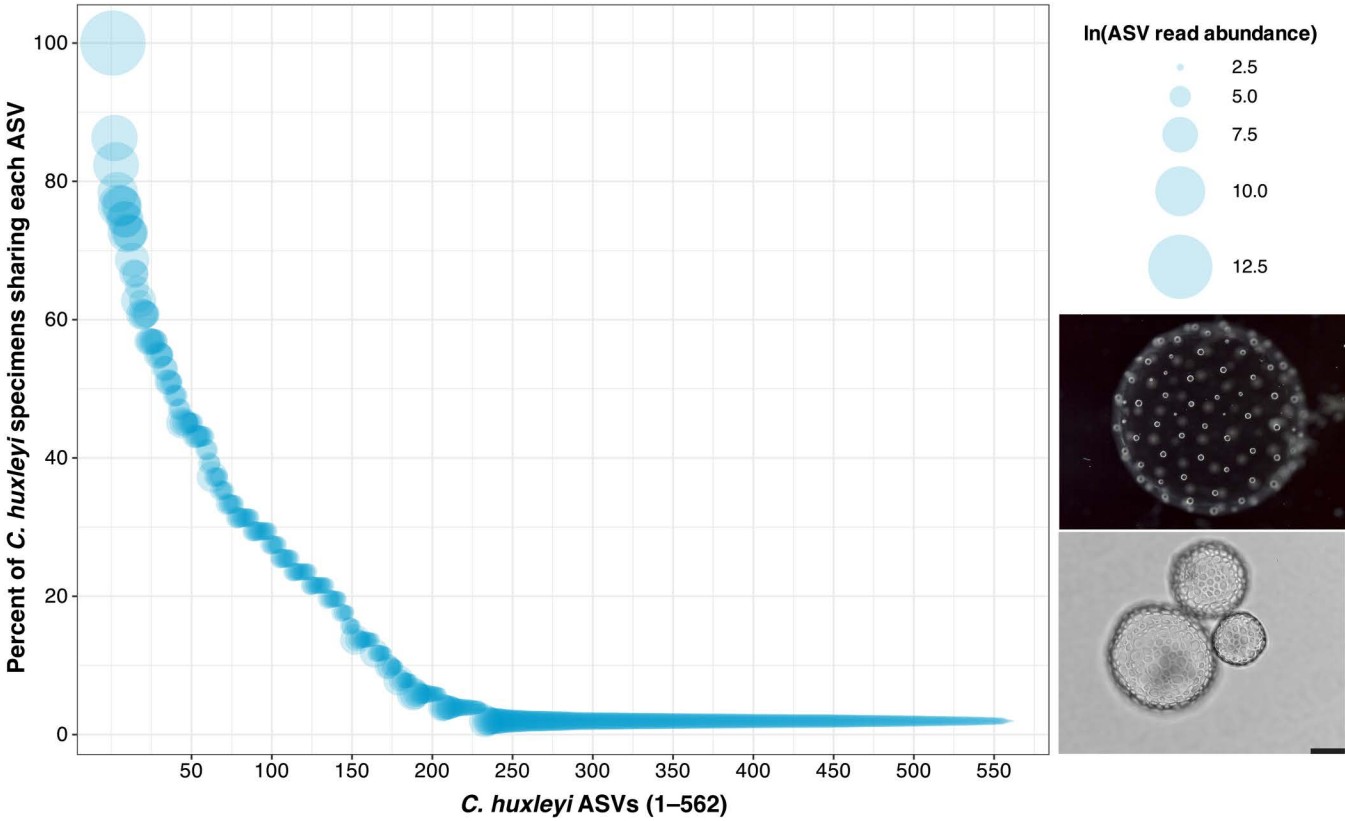

**Fig 3. Intraspecies genetic variation in *Collosphaera huxleyi*.** Each bubble denotes one ASV; the size of the bubble reflects the total read abundance of that ASV across all *C. huxleyi* specimens collected in this study (natural log). The position of bubbles along the y-axis represents the percent of *C. huxleyi* specimens that contained a given ASV (with ≥3 reads). Only one ASV was shared by 100% of specimens (top left), and this ASV also had the greatest number of total reads. ASV bubbles are colored with semi-transparent blue fill; the less transparent the blue color appears, the more ASVs were shared by a similar percent of *C. huxleyi* specimens. The solid blue region at the bottom of the plot reflects the high number of ASVs present in only one specimen (330 ASVs) and their wide-ranging read abundances within those specimens (bubble sizes). Embedded photos show a typical live colony specimen (top) and skeletal remains of three cells from a colony (bottom; scale bar = 50µm). The data underlying this plot can be found in S1 Data, and visualized as a heatmap in S2 Fig.

shared ASVs at all, suggesting separate genetic subgroups that may reflect cryptic species (Table 1). Polycystine morphospecies therefore cannot be easily delimited based on their 18S rDNA sequences alone, because each one contains uncharted genetic richness and an unknown number of genetic subgroups with different ASV compositions.

One way to begin to untangle the degree of genetic differentiation in polycystine morphospecies is to compare the genetic distance within species to the genetic distance between species. Characterizing these genetic distance thresholds could provide an approach for estimating species richness in environmental metabarcoding studies, when morphological species counts are not possible. We quantified radiolarian intragenomic, intraspecies, and interspecies genetic variation by computing mean p-distances (calculated in MEGA 12 [50]). P-distance is the proportion of sites with non-matching bases across all positions of two aligned sequences. Mean p-distance provides a metric of the average degree of difference between all sequence pairs comprising a pre-defined group (e.g., between all ASVs within a specimen or a morphospecies) [50]. Both full-length ASV data and the extracted V4 hypervariable region sequence variants were used to compute intragenomic, intraspecies, and interspecies p-distances (Fig 7a and 7b, respectively).

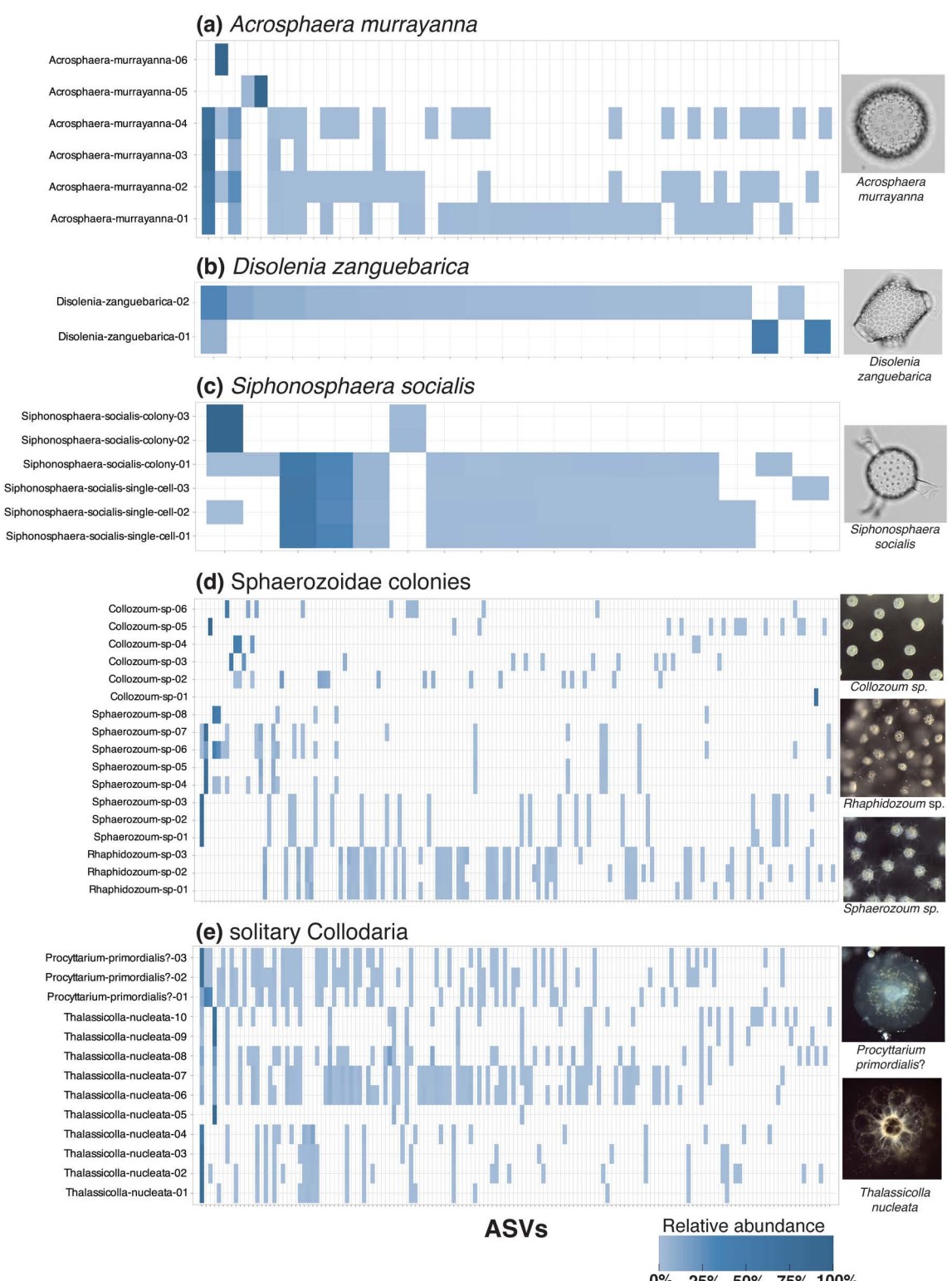

**Fig 4. Heatmaps illustrating intraspecies and intragenomic ASV patterns in Collodaria.** ASV presence and relative read abundance are shown for each morphospecies or group of morphotaxa examined in this study: *Acrosphaera murrayanna* **(a)**; *Disolenia zanguebarica* **(b)**; *Siphonosphaera socialis* **(c)**; Sphaerozoidae colonies **(d)**; and solitary Collodaria **(e)**. Morphological classifications of each specimen are shown on the y-axis of each heatmap

with a random numeric identifier. The x-axis of each heatmap denotes the ASVs (represented by tick marks) present in each specimen of a morphospecies or group of morphotaxa. Relative read abundances of ASVs per specimen are indicated by the shade of blue (light = low abundance, dark = high abundance); ASVs not present in a particular specimen are indicated by white background. Representative micrographs of morphotaxa are shown on the right of each heatmap. All data underlying this figure can be found in S1 Data. Photos of every specimen can be found in S1 Fig.

Results showed that there is indeed greater genetic difference between morphospecies than within them, but the typical p-distances separating intragenomic and intraspecific from interspecific genetic diversity varied widely between taxonomic groups (Fig 7). Among Collodaria, intragenomic variation was expressed by p-distances ranging from 0–0.176 (median: 0.009); Spumellaria had a comparable median intragenomic distance of 0.009 but lower range in values (0–0.037), while Nassellaria specimens showed the lowest median and range (median: 0.001; range: 0–0.032). Intraspecies genetic variation was higher than intragenomic distances for all orders, with median p-distance values of 0.037 for Collodaria, 0.004 for Nassellaria, and 0.013 for Spumellaria. However, orders displayed very different magnitudes of intragenomic and intraspecies genetic variation, with collodarian intragenomic distances on-par with the intraspecies distances in Spumellaria and even surpassing intraspecies distances in Nassellaria. Similarly, interspecies genetic variation was higher than intraspecies variation for all orders, but p-distance values were substantially different between them (Fig 7). The median p-distance separating Collodaria morphospecies in this study was 0.203 (range: 0.083–0.222), 0.113 (range: 0.026–0.164) for Nassellaria morphospecies, and 0.027 (range: 0.011–0.141) for Spumellaria morphospecies. When these analyses were performed using the extracted V4-region ASVs (Fig 7b), the same patterns held but p-distances were greater at each hierarchical level. This is presumably because the relatively short (~440 bp) V4 hypervariable region is expected to contain a higher proportion of site differences between ASVs at each hierarchical group level compared the full-length (~1450 bp) ASVs, which are comprised of both conserved and hypervariable regions. These results indicate that p-distances have promise for delimiting polycystine species-level taxa in metabarcoding studies, but appropriate p-distance thresholds vary considerably based on the length and characteristics of the target amplicon regions, and the order-level classification of the ASVs in question. Additional single-specimen sequencing efforts involving more Radiolaria from different taxonomic groups will improve our ability to define p-distance thresholds for estimating species richness with metabarcoding data.

These findings highlight the challenge of detecting specific morphospecies of interest in DNA metabarcoding surveys using the existing structure of reference sequence databases (i.e., PR2 [4]). Reference sequences derived from isolated specimens of known protistan species are vital for interpreting genetic diversity data. But these individual reference sequences will, in many cases, fail to capture the range of prevalent sequence variants produced by species in the environment. This can lead to overestimation of species richness and evenness in a sample when interpreting molecular data relative to morphospecies data. If we wish to detect a given morphospecies in a DNA survey, it would be more appropriate to organize reference sequence databases by groups of multiple sequence variants that co-occur within the same specimen, and within specimens of the same morphospecies, thereby explicitly accounting for intraspecies genetic variation. Conversely, an approach that is less rigidly bound by morphospecies concepts may ultimately provide us with a more natural characterization of protistan richness. Work on Foraminifera has shown that cryptic species often occupy narrower niches than the morphospecies to which they belong [51]. This means we risk overestimating species' biogeographic ranges and niche breadths if we fail to consider their genetic variation and look only through the lens of morphology. Ecological models predicting future geographic shifts in plankton communities could thus be improved by knowledge of intraspecific genetic variation. Such models utilize species' presumed habitat preferences to predict geographic shifts in plankton communities (e.g., [52]), but if genetically distinct subgroups of some morphologically-described species have different habitat preferences, then our assumptions about their future geographic shifts may be inaccurate. Therefore, interpreting taxa as assemblages of multiple shared ASVs, likely with more than one possible assemblage per morphotype, could allow the use of metabarcoding data to shed light on their ecological preferences and roles that might otherwise remain obscured behind classical morphospecies concepts or ungrouped, discrete ASVs.

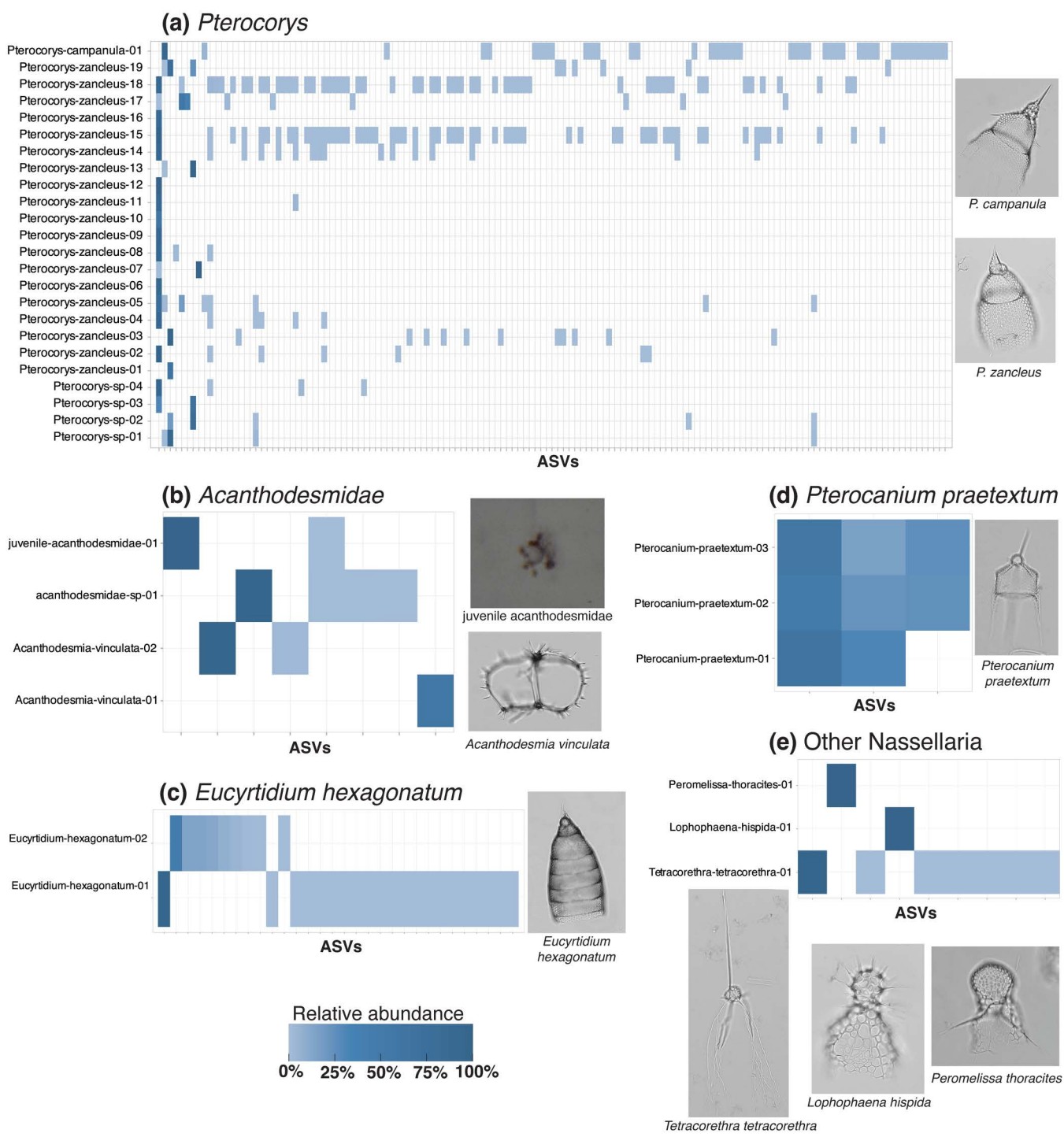

**Fig 5. Heatmaps illustrating intraspecies and intraspecimen ASV patterns in Nassellaria.** ASV presence and relative read abundance are shown for each group of morphospecies and genera: *Pterocorys* **(a)**, *Acanthodesmidae* **(b)**; *Eucyrtidium hexagonatum* **(c)**; *Pterocanium praetextum* **(d)**; and other Nassellaria **(e)**. Morphological classifications of each specimen are shown on the y-axis of each heatmap with a random numeric identifier. The juvenile specimen could not be identified based on morphology; instead, it was classified as family Acanthodesmidae based on its dominant ASV match to a reference sequence (99.9% confidence). The x-axis of each heatmap denotes the ASVs (represented by tick marks) present in each specimen of a morphospecies or group of morphotaxa. Relative read abundances of ASVs per specimen are indicated by the shade of blue (light = low abundance, dark = high abundance); ASVs not present in a particular specimen are indicated as white background. Representative micrographs of morphotaxa are shown on the right and below each heatmap. All data underlying this figure can be found in S1 Data. Photos of every specimen can be found in S1 Fig.

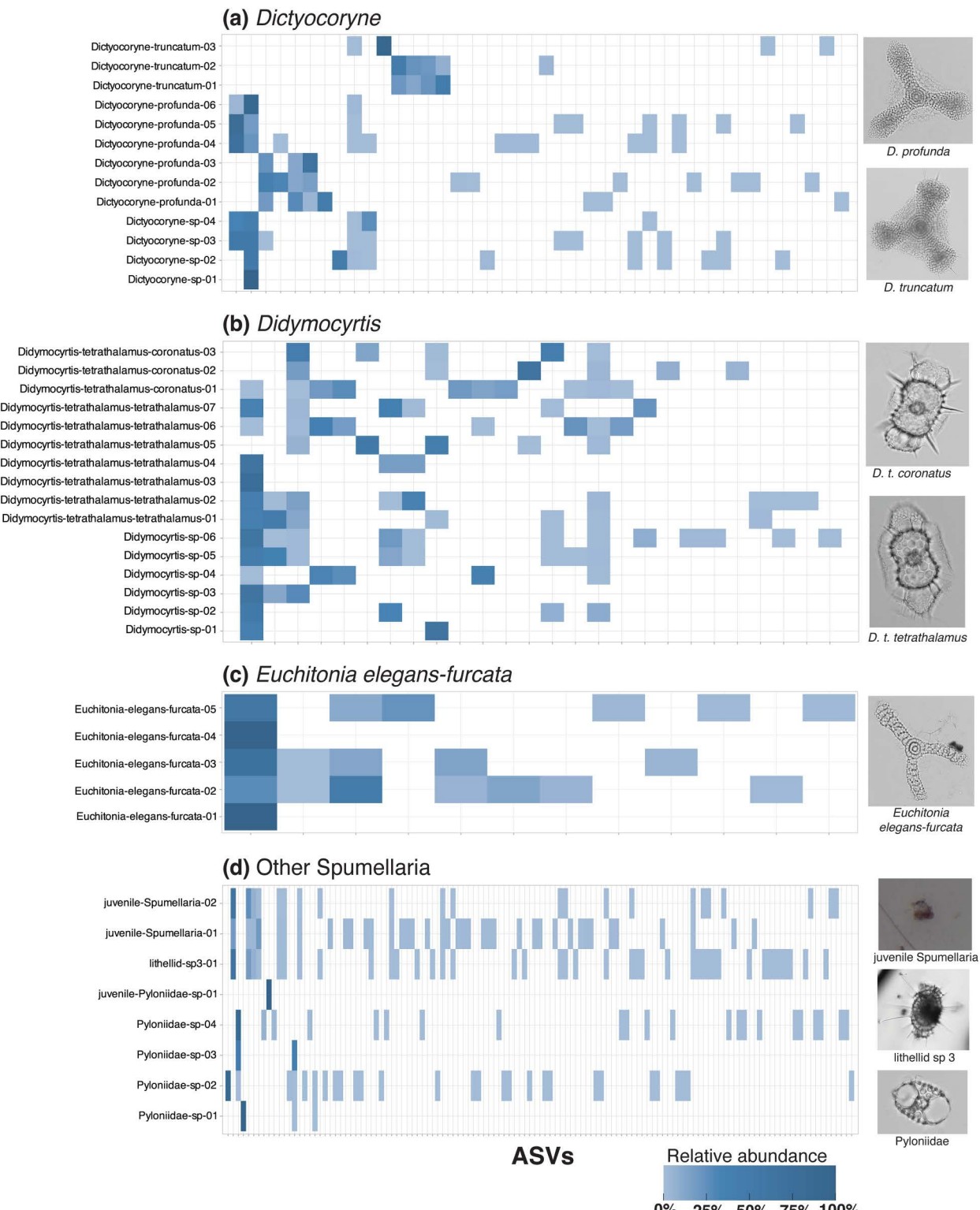

**Fig 6. Heatmaps illustrating intraspecies and intraspecimen ASV patterns in Spumellaria.** ASV presence and relative read abundance are shown for each group of morphospecies or morphotaxa: *Dictyocoryne* **(a)**; *Didymocyrtis* **(b)**; *Euchitonia elegans-furcata* **(c)**; and other Spumellaria **(d)**. Morphological classifications of each specimen are shown on the y-axis of each heatmap with a random numeric identifier. Juvenile specimens were classified

as Spumellaria and Pyloniidae based on the identity of their matching reference sequences (>99% confidence). The x-axis of each heatmap denotes the ASVs (represented by tick marks) present in each specimen of a morphospecies or group of morphotaxa. Relative read abundances of ASVs per specimen are indicated by the shade of blue (light = low abundance, dark = high abundance); ASVs not present in a particular specimen are indicated as white background. Representative micrographs of morphotaxa are shown on the right of each heatmap. All data underlying this figure can be found in S1 Data. Photos of every specimen can be found in S1 Fig.

Intraspecies genetic variation has clear implications for the study of modern and future protistan biodiversity and biogeography, but it can also reshape our understanding of their fossil record, including historical responses to climate change. In paleontological investigations genetic information is typically not available, and morphology is the primary data used to characterize protists' evolutionary and ecological responses to past environmental shifts (e.g., [53,54,55]). If morphologically-defined species contain differing degrees of genetic variation, possibly including cryptic species, then the impact of some morphospecies extinction or origination events in the fossil record could have had broader ecological significance than previously believed. As intraspecies variation becomes better understood for each morpho-taxonomic group, we may be able to better estimate the impacts of macroevolutionary events throughout the group's history.

### Insights gained from morpho-molecular relationships

Long-read metabarcoding provides a unique opportunity to examine both biodiversity and phylogenetic relationships with the same dataset [56]. Most metabarcoding studies utilize short reads of ~300–500 base pairs, making them heavily reliant on matches to full-length reference sequences for interpretation of evolutionary relationships between taxa. Because the length of our reads spanned most of the 18S rRNA gene (~1450 base pairs), we were able to construct a phylogeny from our data that directly informs on evolutionary relationships between ASVs and is decoupled from reference sequence similarity (S4 Fig). In this way, we investigated the connection between morphological taxonomy, total ASV composition, and the phylogenetic relationships of specimens' dominant ASVs, similar to the approach used by others [24,25,26]. Our results variously revealed that genetic information can in some cases: (1) support morphospecies concepts, (2) indicate taxonomic concepts in need of revision, or (3) potentially inform on radiolarians' poorly-understood life stages.

### (1) Supporting morphospecies concepts.

One example of morphospecies supported by genetic information was found in the Spumellaria genus, *Dictyocoryne*. Although *D. profunda* and *D. truncatum* have similar morphological characters (Fig 6a), our results showed that they are indeed genetically distinct. All of the dominant ASVs in *D. truncatum* specimens branched separately from any of the dominant ASVs in *D. profunda* specimens (S4 Fig). Furthermore, none of the dominant ASVs found in the *D. truncatum* specimens were present in any *D. profunda* specimens, and vice versa (Fig 6a). This finding suggests that ASV data can be used to successfully distinguish them in a metabarcoding survey. The specimens assigned to *Dictyocoryne* sp. had only partially-developed skeletons (either due to breakage or incomplete growth), making it difficult to determine their species identity from morphology (S1 Fig). Based on their patterns of shared ASVs and phylogenetic clustering, however, we were able to infer that *Dictyocoryne* sp. specimens #1–4 belong to *D. profunda* rather than *D. truncatum* (Fig 6a; S4 Fig). This exemplifies a case in which molecular data can be used to assist in species classification, even when morphological characters cannot be observed. There was one ASV present in very low abundance in *Dictyocoryne-truncatum*-03 (5 reads) that was also present as a subdominant ASV in three *D. profunda* specimens. However, due to the extremely low number of reads in *Dictyocoryne-truncatum*-03, we interpret this ASV as environmental contamination that was inadvertently sequenced (see "Materials and methods"), rather than an authentically shared ASV between the two *Dictyocoryne* species.

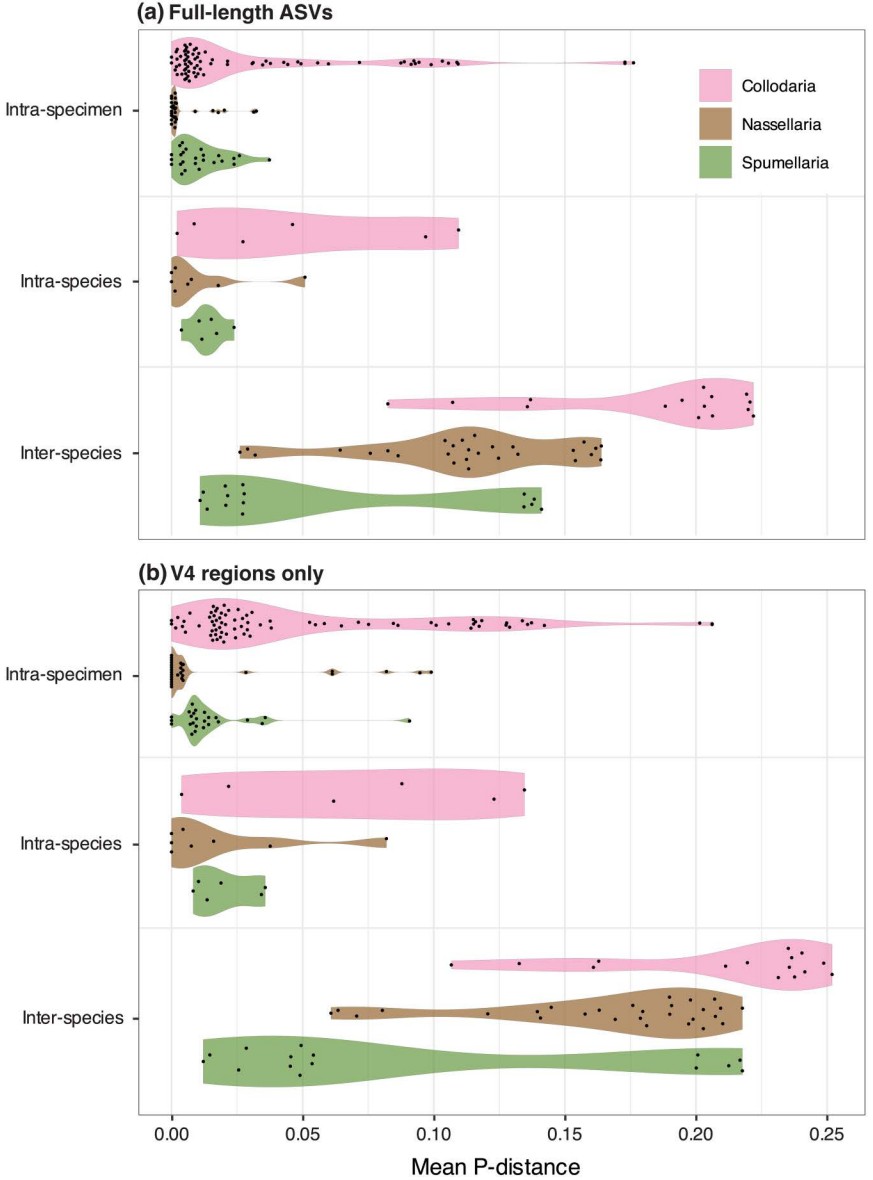

**Fig 7. Mean p-distances of ASVs within individual specimens and morphospecies, and between morphospecies of the same taxonomic order.** Panel **(a)** shows the average intragenomic, intraspecies, and interspecies p-distances for each taxonomic order, based on full-length ASV compositions. Results of the same analyses are shown for the extracted V4-region sequences in panel **(b)**. Only the ASVs derived from specimens morphologically-classified to the species level were used in this analyses. All data underlying this figure is given in S2 Data.

**(2) Highlighting taxonomic concepts in need of revision.**

Results from specimens in the Sphaerozoidae (Collodaria) clearly indicated that a taxonomic revision is needed at the genus level. Each of the Sphaerozoidae genera, *Collozoum* (no spicules), *Rhaphidozoum* (simple spicules), and *Sphaerozoum* (branching spicules) were found to be polyphyletic (Fig 8). Their dominant ASVs were scattered throughout the family Sphaerozoidae, indicating that spicule presence and shape may have evolved independently in multiple clades and should not be considered diagnostic characters at the genus level. These results align with the findings of Biard et al.

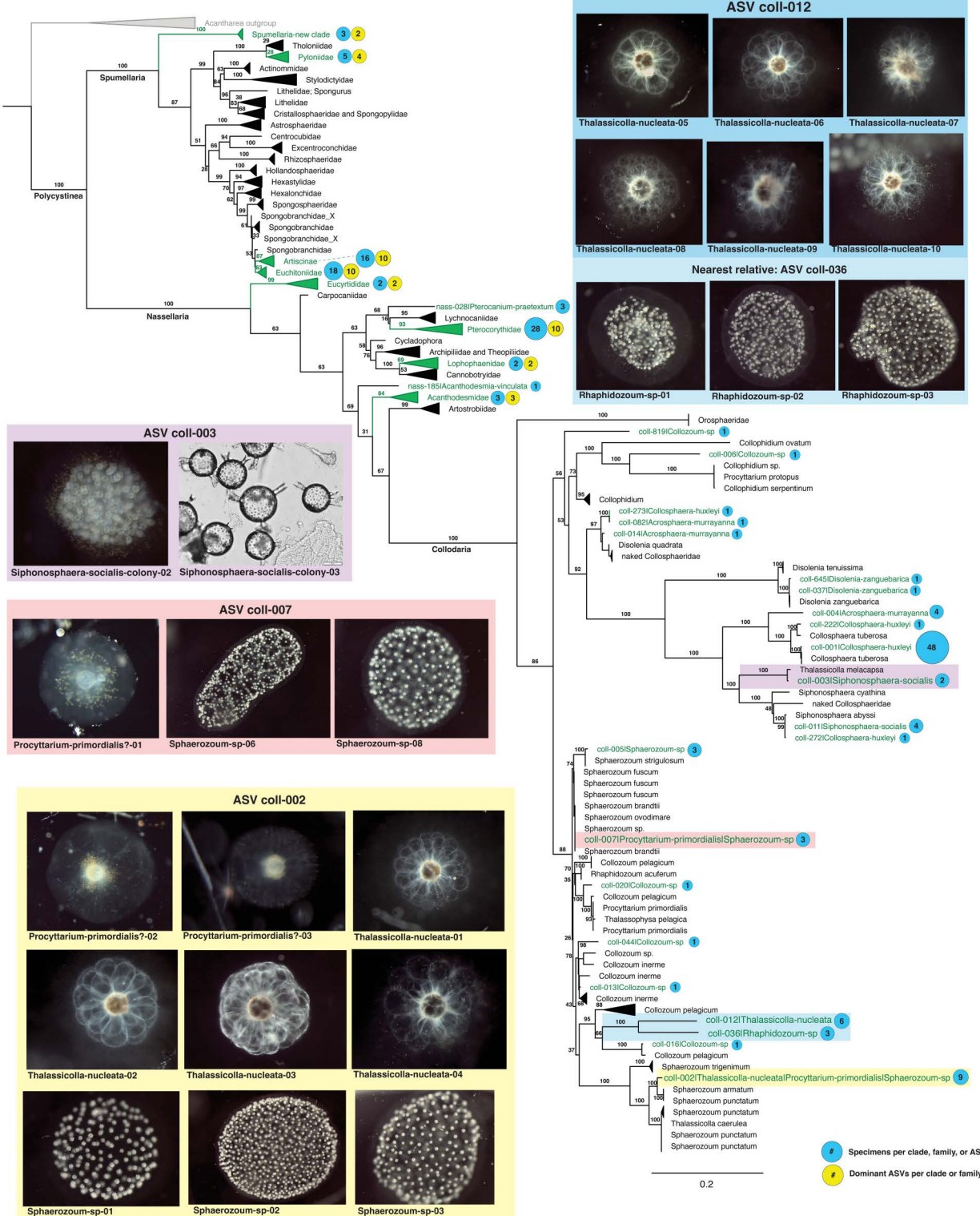

**Fig 8. Phylogenetic context of ASVs generated during this study.** PR2 reference sequences and clades are shown in black (includes a total of 258 polycystine sequences and 18 Acantharea outgroup sequences). The dominant ASVs from specimens in this study are shown in green text with a numeric identifier and the morphology-based taxonomic name. Families or clades in which any of our specimens' dominant ASVs clustered are

highlighted in green and collapsed for clarity. The number of dominant ASVs clustering in each group is shown with yellow bubbles and the number of specimens per clade, family, and dominant ASV is denoted with blue bubbles. Colored boxes headlined with ASV identifiers depict photographs of all specimens that shared the same dominant ASV, and correspond to the colored highlighting in the expanded Collodaria portion of the phylogenetic tree. These cases point to examples in which solitary and colonial Collodaria shared the same dominant sequence variant, or in the case of coll-012, a closely related ASV. Such examples are evidence that solitary and colonial specimens, previously described as different morphospecies, may actually represent different life stages of the same taxon. A full version of this phylogenetic tree that depicts all specimens' dominant ASVs can be found in S4 Fig. Sequence data underlying this tree can be found in S1 Data and S1 Table, and tree parameters are detailed in "Materials and methods".

[24], who also showed that *Collozoum* and *Sphaerozoum* were polyphyletic, but we speculate that *Rhaphidozoum* may be polyphyletic as well. ASVs from our *Rhaphidozoum* specimens did not group in the same clade as the *Rhaphidozoum acuferum* reference sequences (see Fig 8); however, since only two specimens were collected during this study and few reference sequences were available, more data from this genus is needed to confirm its polyphyly.

Our results also suggest that morphospecies concepts of solitary Collodaria may be invalid and instead synonymous with various colonial taxa. Phylogenetic analysis indicated that all solitary Collodaria specimens in this study (morphologically assigned to *Thalassicolla nucleata* and *Procyttarium primordialis*?) either shared their dominant ASV with a colonial specimen, or their closest phylogenetic relative was a colonial specimen (Fig 8). One ASV (coll-002) was dominant in 9 specimens, morphologically-classified as *Sphaerozoum* sp. (colonial), *Procyttarium primordialis*? (solitary), or *Thalassicolla nucleata* (solitary). Because this ASV comprised between 54% and 87% of reads in these specimens, it is not likely the result of environmental contamination or technical error and probably indicates an authentic shared ASV. Additionally, these specimens were picked from three different net tow samples on North Pacific Subtropical Gyre cruise HOT 339, discounting the possibility that their shared dominant ASV is attributable to physical contact during sample collection. ASV coll-007 was also shared by both solitary and colonial taxa, and ASV coll-012 was dominant in several *Thalassicolla nucleata* (solitary) specimens while also being most closely phylogenetically related to the dominant ASV in colonial *Rhaphidozoum* specimens (Fig 8). We additionally found that the dominant ASV in two *Sphonosphaera socialis* colonies (coll-003) was taxonomically assigned to the *Thalassicolla melacapsa* (solitary) reference sequence in PR2 with 99.5% confidence, suggesting these morphospecies may be conspecific. Furthermore, the ASVs mentioned above did not form a monophyletic clade, which is consistent with the findings of Biard et al. [24] that Thalassicollidae is polyphyletic (Fig 8). Together, these results support the hypothesis that solitary Collodaria are conspecific with colonial taxa [24,57,58], rather than a distinct taxonomic group, as they have previously been considered (e.g., [21,22]). Solitary Collodaria should thus no longer be considered valid species in biodiversity surveys, but instead a morphotype that occurs across multiple clades.

(3) **Informing on radiolarian life stages and development.**

Our results indicate that some juvenile polycystine specimens lacking skeletal development may be identifiable to the family level or lower based on their ASV composition and phylogenetic clustering. These specimens were devoid of diagnostic taxonomic characteristics under microscopy, so they may only be distinguishable and possible to classify with sequencing methods. One putative juvenile specimen was genetically classified to the family Acanthodesmidae based on genetic similarity, but due to the poor representation of this family in reference sequence databases classification at lower taxonomic levels was not possible (S5 Fig; Fig 5e; S1 Data). Another putative juvenile was assigned to the family Pyloniidae, due to its genetic match to *Larcopyle butschlii* reference sequences (S5 Fig; Fig 6d). There were two other putative juvenile specimens for which all ASVs were assigned to Spumellaria, but no genetic matches were found below the order level (S1 Data). These juvenile specimens shared their dominant and many subdominant ASVs (19 ASVs shared in total) with a mature spumellarian specimen that was morphologically classified as lithellid sp. 3 (*sensu* Trubovitz et al. [55]) (Fig 6d; S5 Fig). Therefore, all three specimens likely belong to the same open-nomenclature taxon which, based on its phylogenetic clustering, may comprise a novel Spumellaria clade (S4 Fig).

Another key finding from the ASV composition of these juvenile radiolarians is that they suggest no change in intragenomic richness occurs during ontogeny. The juvenile Acanthodesmidae specimen yielded 2 ASVs, which is within the range of ASV richness found in mature specimens morphologically assigned to the same family (*Acanthodesmida-vinculata*-01, *Acanthodesmida-vinculata*-02, and acanthodesmid-sp-01; 1, 2, and 4 ASVs, respectively; Fig 5b). Similarly, the mature Spumellaria specimen (classified as lithellid sp. 3) yielded 40 ASVs, while the two juveniles putatively belonging to the same taxon yielded 39 and 22 ASVs each. Thus, mature specimens of both Nassellaria and Spumellaria appear to have intragenomic variation that is consistent with juvenile individuals, making it improbable that radiolarians accumulate intragenomic variation throughout their life cycles, as ciliates do [59]. This result means that juvenile specimens may be successfully classified based on their ASV composition in precisely the same way as fully-mature specimens in metabarcoding surveys. Metabarcoding surveys therefore have the potential to reveal a more complete picture of radiolarian community composition than microscopy counts or in-situ imaging due to their inherent inclusion of juvenile specimens that are too indistinct to be morphologically identified (see S5 Fig).

Our results also suggest that Collodaria do not accumulate genetic diversity during cell division and colony growth, and different life stages yield comparable genetic signatures. Collodaria colonies have similar magnitudes of intragenomic richness as the individual cells which comprise them, as well as specimens with a solitary life habit. *Siphonosphaera socialis* was sampled as whole colonies of 10s–100s of cells (3 specimens) and also as individual cells picked from different colonies (3 specimens). Single-cells yielded a mean of 12.3 ASVs (sd = 0.6), whereas entire colonies yielded fewer ASVs on average (mean = 6) but more variation in ASV richness (sd = 7). Likewise, Collodaria specimens with solitary versus colonial life habits did not show consistently different magnitudes of intragenomic richness. Solitary collodarians yielded a mean of 35 ASVs each, whereas colonial specimens yielded between 16 and 50 ASVs on average (depending on whether the exceptionally-diverse *Collosphaera huxleyi* was excluded or included, respectively) (Fig 4; S1 Data). We interpret this as support for the hypothesis that Collodaria grow their colonies via binary fission and are made up of individuals that are genetically identical to the originating cell (e.g., [60]). Thus, the genetic richness observed in Collodaria does not appear to be directly related to the number of cells making up their colonies.

Sequence information also has importance for understanding the roles of naked (un-skeletonized) solitary morphotypes, which may be analogous across different polycystine groups. Due to the physical resemblance of two *Procyttarium primordialis*? (Collodaria) specimens with the putative juvenile Nassellaria and Spumellaria specimens we observed (S5 Fig), it is possible that this morphotype is common to all polycystine orders during an early juvenile life stage, and not only in the Collodaria. These specimens' ASV signatures placed them across all three polycystine orders; however, the specimens were morphologically indistinguishable. They shared a light, semi-transparent central spherical feature surrounded by algal symbiont cells encased in a gelatinous cytoplasm that could appear clear or slightly striated or cloudy. Although each cell contained a collection of symbionts, there was no siliceous skeletal development in any of them. To our knowledge, no such morphotype has previously been described in the literature, and we hypothesize that it could be an intermediate stage between swarmer cell and skeletal formation during which the specimen acquires photosymbionts. This finding is consistent with other observations of polycystine ontogeny that document skeletal growth (e.g., [5]), but may represent an even younger life stage prior to initial skeletal formation. If there were distinguishing morphological features among these putative juvenile specimens, they were not obvious under our shipboard dissecting microscope (S5 Fig). It is also possible that some taxa (at least within the Collodaria) have more than one life stage that can be described as a naked solitary cell with symbionts: once early in ontogeny and again during reproduction. Reproductive swarmer cells have been reported in *Thalassicolla* and other solitary collodarian taxa, suggesting that this morphotype could be a reproductive life stage rather than (or in addition to) a juvenile one [12,61]. However, swarmer release has also been reported in colonial Collodaria specimens [62,63], and in fully-skeletonized Nassellaria and Spumellaria [64]. Thus, naked solitary specimens are not the only polycystine morphotype to exhibit evidence of reproductive swarmers. Our findings underscore the need

to examine more such specimens via combined microscopy and nucleic acid sequencing to bridge our current knowledge gaps in radiolarian life cycles and phylogenetic relationships.

## Metabarcoding presents a rich, but enigmatic picture of protistan diversity

Metabarcoding surveys report on the diversity of molecular sequences in a sample or study area (e.g., [7]), but we currently lack a method for relating these molecular signatures to classical morphospecies concepts, which have been the basic unit of much biological research and theory developed over the last two centuries. These molecular species are a new frontier in diversity research and cannot simply be translated into existing morphological, ecological, or biological species concepts [3]. Despite its limitations, marine metabarcoding has already transformed our understanding of protistan life [2]. These data have the power to reveal the behavior, functions, and diversity of microscopic organisms on a staggering scale that has not been approachable using traditional methods.

Although morphospecies are still considered the "gold standard" for measuring protistan diversity, they are ultimately artificial constructs and not necessarily reliable for all groups. For example, it is well-known in the field of radiolarian systematics that some groups are over-split while other groups are over-lumped. Early literature documenting radiolarian species richness (e.g., [65]) included sometimes vague descriptions of newly-described taxa and occasionally lacked any illustration or type material, leading to subsequent confusion of morphospecies categories. Since then, much work has been done to improve radiolarian morphological taxonomic concepts (e.g., [66]) and develop radiolarians into reliable biostratigraphic markers that are frequently employed in the age determination of marine sediments [67]. These marker taxa (chronological species) are typically large (>100 µm) with distinct morphological characters, and their elevated biostratigraphic importance can lead to them receiving great attention to detail (and potential for subspecies-level splitting). Other taxonomic groups have been chronically under-documented, in part due to their irrelevance for biostratigraphy, relatively small size (<100 µm), and complicated morphological characters, leaving many presumed species-level taxa in open nomenclature or in broad species categories (such as the Lophophaenidae [55]). This essentially leaves us with two moving targets: molecular and morphological species concepts, which compounds the difficulty of reconciling them.

Our goal in this long-running reconciliation does not need to be molding one type of data to fit neatly into the other; rather, we can let them work together. Morphological and molecular data are both valuable, as they each can shed light on different aspects of protistan biodiversity, life cycles, and ecological roles. By integrating them we will deepen our overall understanding of these topics in ways that cannot be achieved with morphological or molecular data alone. As we have shown here, it is not a simple task to merge these types of data, due to factors like intragenomic and intraspecies genetic variation, which manifest very differently across taxa. Our results demonstrate that individual polycystine radiolarian specimens usually contain a multitude of ASVs, and their ASV assemblages are inconsistent among specimens belonging to the same morphospecies, even those from the same population. Similar findings of genetic diversity have been observed in other protistan groups as well, suggesting that this phenomenon is widespread, highly complex, and can severely obfuscate diversity metrics in metabarcoding studies (e.g., [39,40]). By adopting a flexible approach to metabarcoding data interpretation that is informed by known characteristics of the target genomes, such as intragenomic variation, copy number, and ploidy, we can begin to build a stronger foundation for new discoveries.

## Materials and methods

### Approach

This study was designed to characterize 18S rRNA gene sequence variability within morphospecies and isolated specimens of polycystine Radiolaria. To accomplish this, live radiolarian specimens belonging to orders Collodaria, Nassellaria, and Spumellaria were collected at sea and processed for both microscopy and long-read nucleic acid circular consensus sequencing. Sequence data were brought through a standard bioinformatic pipeline and customized taxonomic filtering

procedure to identify the exact 18S amplicon sequence variants (ASVs) and their relative read abundances within solitary and colonial specimens belonging to known morphotaxa. ASVs from groups of specimens belonging to the same morphospecies were then used to assess intraspecific genetic variability. A phylogenetic tree was constructed from the long-read ASV data as well as reference sequences, and used in conjunction with ASV presences and abundance patterns to infer relationships between morphotaxa.

## Sample collection

Plankton drift net sampling was conducted at and near Station ALOHA (22° 45'N, 158° 00'W) on Hawaiian Ocean Time-series (HOT) cruises 338 and 339 in July–August 2022 and August–September 2022, respectively. Additional net samples were collected on the Simons Collaboration on Ocean Processes and Ecology-PARticles And Growth in the Oceanic Nutricline (SCOPE-PARAGON II) cruise in August 2022. All cruises took place on the R/V Kilo-Moana.

A SEA-GEAR Model 9000 50 cm diameter plankton net with 50μm mesh was deployed for 5-minute intervals to target solitary Nassellaria and Spumellaria. For the larger Collodaria specimens, a 1m diameter net with 1 mm mesh was used. Samples were collected at a depth of ~5m by-hand with a rope, while a double-release opening-closing mechanism enabled additional net sampling at discrete depths (150m and 300m).

Immediately upon recovery, samples were examined onboard using a dissecting microscope (Leica MZAPO) with 0.63x and 1.6x objectives. Individual polycystine radiolarian specimens were photographed, isolated using a Pasteur pipette, and rinsed 3 times in 0.2μm filtered seawater to remove any attached particles, such as prey or other contaminants introduced during net tow sample collection. Specimens were then transferred to 2 ml cryovials with approximately 0.5 ml of 0.2μm filtered seawater and allowed to incubate at room-temperature for ~4 hours to facilitate prey digestion. Cryovials were then frozen at –80°C onboard and shipped to the University of Southern California on dry ice. In total, 304 isolated specimens were obtained and processed.

## Sample processing and primer design

DNA was extracted from each specimen using the MasterPure Complete DNA and RNA Purification Kit (Biosearch Technologies) following the manufacturer's instructions for "fluid samples." The only modifications were: the debris pellet was retained for taxonomic identification of the specimen's skeletal remains, and DNA from solitary specimens (i.e., single-cells) was resuspended in 20uL rather than 35uL TE buffer in order to increase DNA concentration for PCR amplification (Steps 3 and 8 in Part D of the manufacturer's instructions, respectively). The extracted DNA concentration of each specimen was measured with a Qubit 2.0 fluorometer.

Extracted DNA was PCR amplified using three custom primer pairs designed to target ~1450 bp segments of the 18S rRNA gene in either the Nassellaria, Spumellaria, or Collodaria. Targeted primers (Table 2) were developed by manually inspecting all curated polycystine reference sequences available in the PR2 database v4.14 as well as a long-read environmental DNA sequence database generated by Jamy et al. [56] Using AliView software [68], radiolarian sequences were visually compared to 80% consensus sequences of all other eukaryotic groups in PR2 (at approximately phylum-class level; PR2 taxonomy level 4) to check the specificity of potential primer sequences. Then, primer pair candidates were tested *in-silico* using Silva TestPrime 1.0 and the primer test function in PR2 to confirm that relatively few

**Table 2. Polycystine radiolarian primers designed for this study.**

| Target group | Forward primer (5'-3') | Reverse primer (5'-3') |
|---|---|---|
| Nassellaria | GTCGGTAGTGTATTGGRCTAC | CYGCAGGTTCACCTACGG |
| Spumellaria | GACGTKTCATTCAAATTTCTGCC | GTTACGACTTCTCCTTCCTC |
| Collodaria | GGTTGABCCTGCCAGTAG | CYTATTGTAGCCCGTGCGC |

non-target taxa would be amplified as PCR products. These tests found that with zero mismatches, the primer pair hits were 96% Nassellaria, 76–88% Spumellaria, and 100% Collodaria, indicating that they would theoretically amplify the target taxa with limited contamination from other groups.

Extracted DNA of each specimen was amplified using the primer pair designed for the appropriate radiolarian order, based on initial taxonomic assignments during shipboard collection and inspection. Although Collodaria are phylogenetically positioned within Nassellaria, it is common practice among radiolarian workers to consider them a distinct order-level group due to their unique characteristics among polycystines: colonial life habit, all described species have photosymbionts, sometimes cells contain multiple nuclei, and some taxa lack a silicified skeleton [5]. The Collodaria primer set was initially designed to capture family Collosphaeridae and limit amplification of the other collodarian groups (because Collosphaeridae is the only family in which all described species build skeletons that can be morphologically identified). However, in practice, these primers amplified specimens belonging to all collodarian families. Primer sequences were obtained from Integrated DNA Technologies (IDT) with M13 tails and a 5' block, as recommended by Pacific Biosciences (PacBio) for multiplexed amplicon library preparation (PacBio publication# 101-921-300, revision 2, April 2022). M13 barcodes were also obtained from IDT to allow for asymmetrically-barcoded samples (up to 384 combinations) compatible with PacBio SMRTbell adapter ligation.

Target amplification (PCR-1) was conducted in 25 µL reactions, which included: 12.5 µL KAPA HiFi HotStart ReadyMix, 2.5 µL forward primer (3µM concentration), 2.5 µL reverse primer (3µM concentration), up to 7.5 µL DNA sample, and 7.5 µL–DNA sample volume of molecular biology grade water. Whenever possible, 10ng of DNA was input into the PCR reaction per individual specimen. This amount was typically only attained for colonial specimens, however, as most solitary specimens yielded very low DNA concentrations. Nassellaria and Spumellaria specimens yielded an average of 2.8ng and 3.6ng total DNA, respectively, whereas Collodaria specimens (most of which were colonial) yielded 250ng DNA on average. In cases where a specimen's DNA concentration was < 0.75ng/µL, the maximum volume allowable in the reaction (7.5 µL) was used in PCR-1. Even reactions with ~0.5ng total DNA input often yielded successful PCR results, so low amounts of raw DNA did not appear to severely impact downstream processing. The thermocycler program for PCR-1 proceeded as follows: initial denaturation at 95°C for 5 minutes, then 35 cycles of denaturation at 98°C for 30 seconds, annealing for 45 seconds at 54°C (for Nassellaria and Spumellaria) or 64°C (for Collodaria), extension at 72°C for 90 seconds, followed by a final extension at 72°C for 2 minutes.

Following the first round of PCR, samples were purified with PacBio paramagnetic beads using standard clean-up protocols. Purified PCR-1 products were Qubit-quantified. Specimens that lacked a detectable amount of DNA at this stage were re-run; if they still did not amplify after a second PCR-1 attempt, they were removed from any further processing and analysis. Out of 304 specimens, 54 failed PCR amplification and 18 were lost to a microcentrifuge tube malfunction, leaving a total of 232 successfully-amplified specimens. These samples underwent a second round of PCR (PCR-2) for barcode attachment. The ingredients in PCR-2 were: 12.5 µL KAPA HiFi HotStart ReadyMix, 1.25 µL forward M13 barcode (10µM concentration), 1.25 µL reverse M13 barcode (10µM concentration), up to 10 µL cleaned PCR-1 product, and 10 µL–PCR-1 product of molecular biology grade water. PCR-1 products were diluted as needed so that 10ng of DNA was input into PCR-2. The following thermocycler program was used: initial denaturation at 95°C for 5 minutes, then 2 cycles of denaturation at 98°C for 30 seconds, annealing at 60°C for 30 seconds, and extension at 72°C for 90 seconds, followed by 25 cycles of denaturation at 98°C for 30 seconds, annealing at 66°C for 30 seconds, and extension at 72°C for 90 seconds, with a final extension at 72°C for 5 minutes.

A second PacBio bead clean-up was performed prior to Qubit quantification. PCR-2 products were run on SYBR-safe stained 1% agarose gels to confirm amplification in the size range of the targeted amplicon (~1450 bp) and cleanup of debris fragments were successful. Sample concentrations were standardized and then pooled. The pool was brought to the University of California, Irvine Genomics Research and Technology Hub, where SMRTbell library preparation was performed by staff using PacBio's SMRTbell 3.0 prep kit. The finished library was sequenced on a single PacBio Sequel II SMRT Cell 8M.

The retained skeletons post-DNA-extraction were gently rinsed in deionized water before being placed onto individual glass coverslips. Once moisture had fully evaporated, coverslips were mounted onto glass microscope slides with Canada Balsam in xylene (Refractive Index 1.52). Specimens were observed under transmitted light (Olympus BX51 compound microscope at 100x, 200x, and 400x) so that skeletal remains could be classified to the most precise taxonomic level possible. All taxonomic classifications were performed by ST based on formal descriptions and illustrations in several published literature sources, including: [11,24,38,49,55,57,60,62,65,67]. The curated online taxonomic resource Mikrotax (www.mikrotax.org [69]) was also consulted to confirm species identifications. If specimens lacked a skeleton (i.e., naked Collodaria or juveniles) or if a skeleton could not be identified due to breakage, taxonomic classification was made to the lowest level possible based on initial shipboard photos. The Collodaria specimens in this study were morphologically identified as *Acrosphaera murrayanna* (6 specimens), *Collosphaera huxleyi* (51), *Collozoum* sp. (6), *Disolenia zanguebarica* (2), *Siphonosphera socialis* (6)*, Procyttarium primordialis*? (3), *Rhaphidozoum* sp. (3), *Sphaerozoum* sp. (8), or *Thalassicolla nucleata* (10). Nassellaria specimens were identified as acanthodesmid sp. (1 specimen), *Acanthodesmia vinculata* (2), *Eucyrtidium hexagonatum* (2), *Lophophaena hispida* (1), *Peromelissa thoracites* (1), *Pterocanium praetextum* (3), *Pterocorys campanula* (1)*, Pterocorys* sp*. (4), Pterocorys zancleus* (19), or *Tetracorethra tetracorethra* (1), and one putative juvenile specimen. Spumellaria specimens were comprised of *Dictyocoryne profunda* (6 specimens), *Dictyocoryne* sp. (4), *Dictyocoryne truncatum* (3), *Didymocyrtis* sp. (6), *Didymocyrtis tetrathalamus coronatus* (3), *Didymocyrtis tetrathalamus tetrathalamus* (7), *Euchitonia elegans-furcata* (5), Pyloniidae sp. (4), or lithellid sp. 3 (sensu Trubovitz et al. [55]) (1), and three putative juvenile specimens.

## Data analyses

In total, 5,247,250 barcoded circular consensus reads were received from the University of California, Irvine Genomics Research and Technology Hub, with a mean of 22,617 reads per specimen. These reads were put through the DADA2 bioinformatic pipeline [70] to remove primers, filter by length and quality, denoise errors, call ASVs, and remove chimeras. Sequences were initially checked for primer matches and oriented in the forward direction using cutadapt, with the "linked adapter" setting (-g) and the parameters --revcomp --discard-untrimmed -e 0.1, to account for reverse-complemented reads, remove reads lacking both forward and reverse primer sequences, and limit the allowable error rate to 10%. PacBio Circular Consensus Sequence data must be corrected for orientation because reads can be output in either the forward or the reverse direction. DADA2 removePrimers function was then applied to double-check read orientation using the orient = TRUE parameter. The removePrimers function was also used to confirm that all reads contained primer matches (using the default maximum mismatch threshold of 2), remove any reads that did not contain primer matches, and then trim primer sequences from all passing reads. Next, the DADA2 filterAndTrim function was used to filter reads for quality and length. We used the parameters maxLen = 1600 and minLen = 1300, based on the expected length range of our target amplicons (1400–1500 bp). Following the filtering recommendations for long-read PacBio amplicon data [71], we required that reads contain zero Ns, have a minimum quality score of 3, and a maximum expected error rate of 2 (maxN = 0, minQ = 3, maxEE = 2). Within the DADA2 learnErrors function, we used the parameters "errorEstimationFunction=PacBioErrfun" and "BAND_SIZE=32," as advised for PacBio long-read data [71]. To best detect rare sequence variants that may be shared by multiple specimens, the parameter pool = TRUE was used in executing the dada function. Finally, the removeBimeraDenovo function in DADA2 was used to identify and remove chimeric reads. These steps were designed to separate spurious reads due to technical errors from authentic biological sequence variants [70]. Although we implemented the recommended parameters for quality filtering and calling ASVs with PacBio long-read circular consensus sequence data [71], there is no set of parameters that is guaranteed to remove all PCR and sequencing errors while retaining all authentic biological sequence variants. Thus, it is possible that our stringent filtering methods removed some authentic sequence variants or failed to remove some erroneous reads.

Table 3 tracks the mean percent of reads that passed through each stage of the DADA2 pipeline for each order. It is presumed that the relatively low passing rates at the quality and length filtering step is primarily due to length rather than quality. The sequencing facility reported that 100% of our barcoded reads had Phred quality scores ≥Q20. Thus, PCR artefacts or off-target amplification are the most likely explanations for our passing rates. The mean read length of the raw demultiplexed data was 1252 bp; however, based on known genomic characteristics of the target taxa, we expected on-target reads to be approximately 1400–1500 bp long. We thus required that reads passing our length filter be between 1300 and 1600 bp long, which alone would have removed approximately half the raw reads for being too short. These relatively short reads may have originated from artefacts generated during PCR, or from contaminating taxa that happened to have shorter regions of the 18S gene and were unexpectedly amplified by our primers.

ASV data were then imported into QIIME2 [72] for taxonomic assignment and filtering. In QIIME2, naïve bayes classifiers were trained on the PR2 database with the custom primer pairs (Table 2). After taxonomic classifications were made (using minimum confidence = 0.3), any sequences that did not belong to polycystine Radiolaria or had < 3 total reads were filtered out. In an additional effort to remove potential sources of environmental contamination from each specimen, any ASV that was not assigned to the same family as the morphospecies was excluded from analyses. An exception to this protocol was made for naked Collodaria and putative juvenile specimens, which lacked the skeletons needed for confident morphospecies identifications. For these specimens, ASVs with any polycystine family assignment were permitted. Finally, a per-specimen read abundance threshold was applied, which required each ASV to have at least 3 reads in a given specimen before it was counted as present in that specimen. After these vetting procedures, any specimens lacking a minimum of 5 taxonomically-consistent ASV reads were removed from all analyses.

A maximum-likelihood phylogenetic tree (constructed with RAxML-NG [73]) was used to detect patterns of suspicious topology that may indicate hidden chimeric sequences. Input data included 258 polycystine radiolarian reference sequences, 18 Acantharia outgroup sequences downloaded from the PR2 database (S1 Table), and the most abundant sequence from each of the specimens in this study (available in S1 Data). Sequences were aligned using MAFFT [74] with linsi parameters and then trimmed with a 5% gap threshold using trimAL [75]. The tree utilized the GTR + G evolutionary model with 100 bootstrap replicates and 1853 aligned positions. After visual inspection, ASVs that formed long-branching clades without any reference sequences and which had relatively low read numbers (i.e., < 50) were manually checked for chimeras by querying NCBI–BLAST (https://blast.ncbi.nlm.nih.gov/Blast.cgi) with the first ~300 bases and last ~300 bases of each sequence separately. This test led to the discovery of 14 chimeric ASVs that had not been removed by the DADA2 algorithm and were removed manually. All these chimeric sequences had been taxonomically assigned to family Pterocorythidae (Nassellaria) in PR2, although NCBI–BLAST identified their sequence fragments as belonging to various groups, including Rhizaria, stramenopiles, and alveolates.

Following all quality checking procedures, 173 out of 232 individual specimens passed through all processing steps (75%), comprising 1,224 unique ASVs among them. This included 836 ASVs assigned to Collodaria (95 specimens), 187 ASVs assigned to Nassellaria (36 specimens), and 201 ASVs assigned to Spumellaria (42 specimens). This made up the dataset used for all analyses in this study (S1 Data). There were 59 specimens that successfully amplified during PCR but did not pass bioinformatic processing and filtering steps,

**Table 3. Mean percent of total input CCS reads that passed through each step of the DADA2 pipeline per polycystine radiolarian order.**

| Order | Primers trimmed (% input) | Quality & length filtered (% input) | Denoised (% input) | Non-chimeric (% input) |
|---|---|---|---|---|
| Nassellaria | 92 | 37 | 33 | 30 |
| Spumellaria | 89 | 35 | 30 | 24 |
| Collodaria | 90 | 59 | 56 | 53 |

due to their lack of sufficient reads (<5 reads) belonging to taxonomically-appropriate ASVs. It is interpreted that any apparent amplification was due to off-target environmental DNA that may have come into contact with the specimen prior to or during sample collection, and the target DNA coincidentally failed to properly extract or amplify from the specimen. A similarly-designed study [28] found that environmental contamination can contribute exogenous sequence data when amplifying isolated radiolarian cells, even after cleaning and prey digestion. Thus, while we made our best efforts to remove contamination from every specimen in this study via stringent sample preparation and bioinformatic filtering by both read abundance and ASV taxonomic assignment (described above), we cannot absolutely guarantee that every ASV we document here originated from within the target specimen.

To compare the genetic richness information provided by our long-read ASVs with commonly used V4 hypervariable region ASVs, we extracted V4 sequences from each full-length ASV. V4 regions were identified by aligning all sequences with MAFFT [74] using lingsi parameters, then searching for matches to the general eukaryotic primers TAR-EukF1 (5'-CCAGCASCYGCGGTAATTCC-3')/ TAR-EukR3 (5'-ACTTTCGTTCTTGATYRA-3') [76] using AliView software [68]. ASVs that had identical V4 regions were merged for the purpose of building V4-only occurrence charts (S1 Data; sheets 4–6) and exploring V4-only intragenomic variation (Fig 1, e–h) and intra- and inter-species variation (Fig 7b). After merging identical V4 sequences, there was a total of 604 unique V4 sequences in the dataset; 417 assigned to Collodaria, 70 to Nassellaria, and 117 to Spumellaria.

We computed p-distances within and between specimens and morphospecies to provide quantitative metrics of intragenomic and intraspecies genetic variation. P-distance is the proportion of sites that are different between two aligned sequences being compared. We used MEGA (Molecular Evolutionary Genetic Analysis) 12 software [50] to calculate the mean p-distances within and between groups of ASVs belonging to individual specimens, and comprising each morphospecies. The MEGA analysis option "Compute within group mean distance" was used to obtain average intragenomic and intraspecies p-distances within each polycystine order, and the option "Compute between group mean distance" provided average genetic distances between specimens and morphospecies (Fig 7; S2 Data). Distance calculations were performed separately on full-length ASVs and V4-region ASVs for comparison. Only specimens that were morphologically-identified to the species level were included in p-distance calculations, to avoid mixing unequal taxonomic levels in the same analysis.

ASV richness, presence, and abundance patterns within specimens, morphospecies, and orders were assessed and visualized in R (v4.3.1, "Beagle Scouts") [77] using the dplyr [78] and ggplot2 [79] packages. ASVs for each of the 173 specimens were ranked based on decreasing relative abundance per specimen in base R. The most abundant ASV in each specimen was used to construct a maximum likelihood phylogenetic tree in RAxML-NG using the same parameters as the initial tree described above, and was visualized in FigTree v1.4.4 [80] (Fig 8; S4 Fig). This phylogenetic tree was used to infer relationships among specimens collected for this study with previously-published radiolarian sequences as a framework. A flowchart summarizing all bioinformatic processing and analytical steps is given in S6 Fig.

## Supporting information

**S1 Fig. Plates of all 173 isolated specimens used in analyses: living and skeletal.** Photos taken of live specimens in darkfield and brightfield with the shipboard dissecting scope have been cropped and adjusted for white balance and contrast when necessary. Photos taken of skeletal remains with the compound microscope are unedited. Specimens collected on the SCOPE-PARAGON II cruise were not photographed live, so only photos of skeletal remains are included. Other specimens that only have one photo either could not be photographed on board the ship due to high rocking motion, or they broke beyond recognition during/after DNA extraction and could only be identified from their live photos. (PDF)

**S2 Fig. Heatmap illustrating ASV presence and abundance in all 51 *Collosphaera huxleyi* specimens analyzed in this study.** Specimen names are shown on the y-axis, corresponding to specimen names in S1 Data and S2 Fig. The x-axis denotes individual ASVs (represented by tick marks) present in each specimen of the morphospecies. Relative read abundances of ASVs per specimen are indicated by the shade of blue (light=low abundance, dark=high abundance); ASVs not present in a particular specimen are indicated as white background. Representative micrographs of *C. huxleyi* are shown below the heatmap.
(PDF)

**S3 Fig. Distribution of relative read abundances among the top 10 ranking V4-ASVs per specimen.** Dots represent relative V4-ASV abundances (percent of total reads yielded by each specimen) at each designated rank (1st–10th) for all 173 specimens. Boxplots denote the median, interquartile range, maximum, and minimum values (outliers excluded) of V4-ASV relative abundances for each order (Collodaria=pink; Nassellaria=brown; Spumellaria=green). Data underlying this figure is in S1 Data.
(PDF)

**S4 Fig. Expanded phylogenetic tree showing the dominant ASVs of all 173 specimens.** Tip labels indicate the morphological taxonomy of specimens with a given dominant ASV, as well as the number of specimens that shared the same dominant ASV. ASV identifiers correspond with those in S1 Data. Polycystine radiolarian reference sequences downloaded from PR2 are denoted with black text. Green text indicates new ASVs obtained from specimens in this study. Parameters used to generate this tree are described in "Materials and methods." Accession numbers for the polycystine reference sequences and acantharian outgroup sequences can be found in S1 Table.
(PDF)

**S5 Fig. Putative juvenile specimen morphologies across all polycystine Radiolaria orders, photographed live under darkfield and brightfield microscopy (shipboard dissecting scope; Leica MZAPO).** Each box displays specimens that shared dominant ASVs and are presumably intraspecific, and/or have a dominant ASV that closely matches a reference sequence in PR2 or GenBank. Background colors indicate the polycystine order determined based on ASV taxonomic assignments (pink=Collodaria; brown=Nassellaria; green=Spumellaria). These solitary specimens lacked any siliceous skeletonization needed for morphological taxonomic classification, but molecular sequencing revealed taxonomic identities in all three orders. This morphotype is therefore hypothesized to be a previously-undescribed juvenile life stage that has similar characteristics in Collodaria, Nassellaria, and Spumellaria.
(PDF)

**S6 Fig. Bioinformatic flowchart, summarizing each step of the data processing and analysis workflow used in this study.** The purple box denotes the type of input data, blue boxes show intermediate steps in the workflow, and green boxes indicate the main results discussed in this study.
(PDF)

**S1 Data. Occurrence chart of all long-read and V4 region ASVs per specimen and their read abundances.** There are individual sheets for Collodaria, Nassellaria, and Spumellaria specimens, titled either "full-length ASVs" or "V4-only" respectively. The first column of each sheet lists the ASV identifiers (as discussed in the text and displayed in figures). The second column gives the full ASV sequence. The third column, "Taxon," provides the taxonomic assignment of each ASV based on PR2 reference taxonomy, and the "Confidence" column lists the confidence level of each molecular taxonomic assignment on a scale from 0–1. Subsequent columns give the specimen morphological taxonomic assignments with numeric identifiers (as referenced throughout in the manuscript text and in figures), and the absolute number of reads of each ASV produced by individual specimens. All ASVs that matched the family

assignment of each morpho-specimen are included, even those with <3 reads per specimen. The sheets containing extracted V4-regions of each ASV indicate which of the original long-read sequences had unique V4 regions versus identical V4 matches with other sequences (causing their read counts to be merged in the occurrence chart). Each merged V4 sequence is labelled "merged" in the ASV_ID column; all other V4 ASVs retain the same IDs as their long-read counterparts. For the analyses in this study, ASVs with <3 reads in each specimen were removed to account for potential contamination (see "Materials and methods"), but the data in raw form is provided here to enable maximum flexibility in future use.
(XLSX)

**S2 Data. P-distances quantifying genetic variation within and between specimens and morphospecies.** The first sheet lists intra-specimen mean p-distances (number of base differences per site averaged across all sequence pairs comprising each specimen). P-distances between the groups of ASVs comprising each morphospecies are provided in the matrices in second sheet of the dataset; calculations within each taxonomic order were computed separately. Likewise, intra-morphospecies and inter-morphospecies p-distances are given in the third and fourth sheets, respectively. When a specimen or morphospecies only yielded a single ASV, p-distance could not be calculated ("N/A" cells in the spreadsheets). Only specimens morphologically-classified to the species level were included in p-distance analyses.
(XLSX)

**S1 Table. List of accession numbers for all PR2 reference sequences included in the phylogenetic trees (Fig 8 and S4 Fig).**
(XLSX)

## Acknowledgments

The authors would like to thank the Hawaiian Ocean Time-series (HOT) program for allowing ST to sail as a guest scientist and collect samples on cruises HOT 338 and HOT 339. We also thank Jennifer Beatty and Samantha Gleich for collecting Collodaria specimens for this study on SCOPE-PARAGON II.

## Author contributions

**Conceptualization:** Sarah Trubovitz, David A. Caron.

**Data curation:** Sarah Trubovitz.

**Formal analysis:** Sarah Trubovitz, Miguel M. Sandin.

**Funding acquisition:** Sarah Trubovitz, David A. Caron.

**Investigation:** Sarah Trubovitz.

**Methodology:** Sarah Trubovitz, Miguel M. Sandin.

**Project administration:** David A. Caron.

**Resources:** David A. Caron.

**Supervision:** David A. Caron.

**Validation:** Miguel M. Sandin.

**Visualization:** Sarah Trubovitz, Miguel M. Sandin.

**Writing – original draft:** Sarah Trubovitz.

**Writing – review & editing:** Sarah Trubovitz, Miguel M. Sandin, David A. Caron.

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
