## [Decision Letter · Decision Letter 0]

PONE-D-24-55509Genetic variability in microbial eukaryotes reshapes marine biodiversity assessment in the age of amplicon sequencingPLOS ONE

Dear Dr. Trubovitz,Thank you for submitting your manuscript to PLOS ONE. I have received evaluations by four experts in the field  - all evaluations were enthusiastic and in support of your work, while providing constructive suggestions to revise the work. I recommend the authors pay attention to the following points raised by the reviewers: 

Analysis of the variable regions of the long-read amplicons to confirm whether the long-read level corresponds to the same variability of the short-read level (e.g., is the v4 region the same across many of these long-read ASVs);Analysis of degree of similarity/differences between ASVs of the same or between different morphospecies;Methodological clarifications of the analytical pipeline (which identification guides used; specific DADA2 parameters; changes to accommodate long read sequencing);Reformatting of Table 1 based on taxonomy rather than alphabetically;Clarification of the cause for low passing rates of CCS reads in the Quality and Length filtering step;Analysis of the proportion of ASVs shared between juvenile and mature radiolarians to support statements of overestimations of species richness in radiolarian metabarcoding surveys; Several of the reviewers made specific suggestions to include specific literature to support and/or discuss your findings.

I am very much looking forward to a revised version of this manuscript.

We look forward to receiving your revised manuscript.

Kind regards,

Claudia Isabella Pogoreutz

Academic Editor

PLOS ONE

Journal requirements: When submitting your revision, we need you to address these additional requirements. 1. Please ensure that your manuscript meets PLOS ONE's style requirements, including those for file naming. The PLOS ONE style templates can be found at https://journals.plos.org/plosone/s/file?id=wjVg/PLOSOne_formatting_sample_main_body.pdf and https://journals.plos.org/plosone/s/file?id=ba62/PLOSOne_formatting_sample_title_authors_affiliations.pdf. 2. Thank you for stating the following financial disclosure:  [This project was supported by the National Science Foundation Postdoctoral Research Fellowship in Biology Award #2109767 (to ST) and the Simons Collaboration on Ocean Processes and Ecology p49802 (to DAC). MMS was supported by a postdoctoral fellowship from the Beatriu de Pinós programme of the Government of Catalonia's Secretariat for Universities and Research of the Ministry of Economy and Knowledge (grant #2021BP00068).].  Please state what role the funders took in the study.  If the funders had no role, please state: ""The funders had no role in study design, data collection and analysis, decision to publish, or preparation of the manuscript."" If this statement is not correct you must amend it as needed. Please include this amended Role of Funder statement in your cover letter; we will change the online submission form on your behalf.

Reviewers' comments:

Reviewer's Responses to Questions

**Comments to the Author**

1. Is the manuscript technically sound, and do the data support the conclusions?

Reviewer #1: Yes

Reviewer #2: Yes

Reviewer #3: Yes

Reviewer #4: Yes

2. Has the statistical analysis been performed appropriately and rigorously? 

Reviewer #1: Yes

Reviewer #2: Yes

Reviewer #3: Yes

Reviewer #4: Yes

3. Have the authors made all data underlying the findings in their manuscript fully available?

Reviewer #1: Yes

Reviewer #2: Yes

Reviewer #3: Yes

Reviewer #4: Yes

4. Is the manuscript presented in an intelligible fashion and written in standard English?

Reviewer #1: Yes

Reviewer #2: Yes

Reviewer #3: Yes

Reviewer #4: Yes

5. Review Comments to the Author

Reviewer #1: General Comments:

Trubovitz et al present a fascinating exploration of the intraspecific and intragenomic variability within polycystine Radiolarians. This work is vital as it allows researchers to connect the vast amount of morphological data available for the clade with the equally as vast genetic data recently generated. The methods are thorough, and the researchers went to great lengths to ensure their sequences were biologically representative (great stringent workflow for filtering reads) and data was sound. One key finding is that intragenomic variability within radiolaria is likely a significant source of the total genetic richness being measured in metabarcoding surveys, therefore these studies should account for this variability within their study groups. In line with this, the ASV diversity within single species was also very interesting to see. A suggestion I have that would further support/confirm the results would be to analyze the variable regions of the long-read amplicons so that you can see if the variability you see at the long-read level corresponds to the same variability at the short-read level. For instance, is the V4 region (the most commonly used in metabarcoding surveys) the same across many of these long-read ASVs? Overall, I feel that this study is suitable for publication after some minor clarifications and this quick V4 analysis. Given my background is predominately in metabarcoding and marine microbial diversity, and not in Radiolaria specifically, I would also suggest that an expert in Radiolaria be consulted before publication.

Introduction:

Line 48- “morphologically-referenced sequence databases”. I am unsure about this definition. PR2, which is the example used, is based on phylogenetics. The sequences are curated based on where they fall in maximum-likelihood trees. Yes, some sequences which originate from cultures have an original morphologically linked ID, but if a phylogenetic tree shows it branching elsewhere, the taxonomic assignment of the sequence will be changed.

Lines 53-54- I’m not so sure that morphology-based taxonomy is the gold standard in protistology. Most work in the last 10-20 years has focused on phylogenomics to denote new lineages of the eukaryotic tree of life and reclassify those that were taxonomically assigned morphologically in the past.

Line 224- This sentence is a little clunky. Please re-phrase.

Methods:

Line 800- Did a single researcher perform the taxonomic classification? Were any other researchers consulted for the morphological classifications? What was the identification guide used?

Line 819- What parameters were used for quality and length filtering in DADA2? Which unique changes were made to the workflow to account for the long-reads?

I would suggest looking into the V4 region (most common region for metabarcoding studies) of these long read ASVs to see if they correspond to the same amount of variability, as this is what would really reveal if the different ASVs you see at the long-read level correspond to the variability at the short read level.

Results & Discussion:

Figure 1- a short title within or above each chart would be useful for data interpretation.

Heatmaps- the light shading for high abundance and dark shading for low abundance seems reversed for me. I would suggest switching these around or changing the color scheme all together to be more distinct (might help to also remove the gray background?)

Beautiful images and vizualization of the data in Fig 7, great job!

Line 605- “not inconsistent” to “consistent”

Reviewer #2: This manuscript entitled " Genetic variability in microbial eukaryotes reshapes marine biodiversity assessment in the age of amplicon sequencin" provided much information and discussion on the intragenomic and intraspecies genetic variation in one group of marine protists, the polycystine Radiolaria. The authors analyzed the combined microscopy and long-read 18S rRNA gene amplicon data sequencing data and made interesting and important findings that will have great implications for the interpretation of metabarcoding data. The manuscript is well-written, the text is well structured, clear and easy to understand. The results are well illustrated and the manuscript is accompanied by a large amount of supplementery material (photos of all 173 isolated specimens used in analyses, heatmaps illustrating ASV presence and abundance in all analyzed specimens, expanded phylogenetic tree based on the dominant ASVs and ect.). The authors have done extensive and important research. This manuscript is helpful to readers and researchers who are working on or beginning the metabarcoding study of marine protists.

My minor comments are addressed below:

Line 193. “…relative abundance (scaled from 0 to 1)…” If I understand correctly, relative abundance is calculated as n%/100%. Please clarify this in the caption to the figure. You use “%” in the text (lines 177-190). The discrepancy between the figure and the description is a bit confusing.

Lines 207-209. … I would suggest to be more specific. I would suggest adding exact copy numbers of the 18S rRNA gene to the phrases... “astoundingly high…”, “…similar …” “…fewer copies…”

Line 252. Table 1. The list is presented in alphabetical order by morphospecies name. I would suggest reformatting this list based on taxonomy (orders Nassellaria, Collodaria, Spumellaria). In my opinion, this way the data will look more structured and will correspond to its description.

It should also be noted that the main objective of this study is to investigate the intragenomic and intraspecies genetic variation in the polycystine Radiolaria and integrating morphological and molecular information. However, in my opinion it would be interesting to estimate the degree of similarity (or the degree of differences) between sequences of different ASVs of one sample/morphospecies and/or between different morphospecies (for example based on p-distans). I recommend this aspect needs to be addressed in the future. It will add useful information and clarity to the study. And it will probably allow us to determine the boundaries of intragenomic and intraspecies genetic variation among polycystine Radiolaria.

Reviewer #3: The article is well written and structured, interpreting complex data with remarkable clarity. The organization of the content ensures a seamless flow, making the information accessible. It demonstrates the high intragenomic variance in polycystine Radiolaria and the importance of incorporating this variability in future metabarcoding survey.

My main concern would be regarding the Table 3: More than half of input CCS reads was filtered out at the Quality & Length filtering step. However, about 89% reads passed after removing primers and filtering in the literature cited (Line 823). Is there any possible cause to the low passing rate? Was the Q score of raw data tested in other quality control tools? It'd be better if an explanation of the low passing rate is included.

Besides it, I only have a few minor comments and suggestions:

Line 29 – 33: It is a very nice summary of results. But it would be nice to expand a little if possible.

Line 66 – 70: I believe there are also other research using metabarcoding as the Tara Oceans expedition and generated similar data. I think you can compare or add a few more references to back it up.

Line 377-383: It’d be more persuasive with some references. Are there previous research discuss the relation between genetic variation and morphological differences in Radiolaria or protists?

Line 512: Is the question mark inside brackets a typo?

Line 557 – 567: In addition to ASV richness, do these juvenile radiolarians share some of ASVs with their mature specimens? If no ASV is shared, it could support the overestimation of species richness in metabarcoding surveys.

Reviewer #4: The paper “Genetic variability in microbial eukaryotes reshapes marine biodiversity assessment in the age of amplicon sequencing” by Turbovitz and co-authors have used a combined microscopy and long-read sequencing approach to address the relationship between molecular data and morphologically defined species of Radiolaria. The authors show that more specimens should be examined via combined microscopy and molecular analyses to understand more about radiolarian life cycles and phylogenetic relationships.

Their results demonstrates that intraspecies and intragenomic variation in radiolarians is a likely reason for the discrepancy between morphological richness and genetic richness, as the average specimen in their dataset yielded 12 unique ASVs. These findings have major implications for diversity interpretations in metabarcoding studies and should be followed up in future studies by protistologists in general, since intragenomic variation might also be high in other protist groups.

The manuscript is very well written, and it was a pleasure to read it. The findings by Turbovitz and co-authors can be interesting for many researchers.

I only have a few minor comments for the authors:

• L.88: Although it is natural to make a selection of papers to cite, I recommend that the authors also include Krabberød et al. 2011 in the list of papers cited here (https://doi.org/10.1371/journal.pone.0023526)

• I believe the correctly spelled version of i.e./e.g. is with a trailing comma (i.e.,/e.g.,)?

• More a question than a comment: I think it would have been interesting to see what happened to the ASVs if these were clustered into OTUs at different levels (99%, 98%, 97% etc.) – maybe something to consider.

6. PLOS authors have the option to publish the peer review history of their article (what does this mean? ). If published, this will include your full peer review and any attached files.

**Do you want your identity to be public for this peer review?** For information about this choice, including consent withdrawal, please see our Privacy Policy .

Reviewer #1: No

Reviewer #2: No

Reviewer #3: No

Reviewer #4: **Yes: ** Marit F. M. Bjorbækmo

---

## [Author Response · Author response to Decision Letter 1]

3 Apr 2025

Reviewer #1:

General Comments:

Trubovitz et al present a fascinating exploration of the intraspecific and intragenomic variability within polycystine Radiolarians. This work is vital as it allows researchers to connect the vast amount of morphological data available for the clade with the equally as vast genetic data recently generated. The methods are thorough, and the researchers went to great lengths to ensure their sequences were biologically representative (great stringent workflow for filtering reads) and data was sound. One key finding is that intragenomic variability within Radiolaria is likely a significant source of the total genetic richness being measured in metabarcoding surveys, therefore these studies should account for this variability within their study groups. In line with this, the ASV diversity within single species was also very interesting to see. A suggestion I have that would further support/confirm the results would be to analyze the variable regions of the long-read amplicons so that you can see if the variability you see at the long-read level corresponds to the same variability at the short-read level. For instance, is the V4 region (the most commonly used in metabarcoding surveys) the same across many of these long-read ASVs? Overall, I feel that this study is suitable for publication after some minor clarifications and this quick V4 analysis. Given my background is predominately in metabarcoding and marine microbial diversity, and not in Radiolaria specifically, I would also suggest that an expert in Radiolaria be consulted before publication.

Response: We are delighted that this Reviewer found the manuscript fascinating and its content vital for tackling the difficult task of collating morphological and sequence diversity within the Radiolaria. We welcome the possibly for a radiolarian expert to look at the manuscript as well, although we suspect that one or more of the other three Reviewers has that expertise, as we authors do.

Introduction:

Line 48- “morphologically-referenced sequence databases”. I am unsure about this definition. PR2, which is the example used, is based on phylogenetics. The sequences are curated based on where they fall in maximum-likelihood trees. Yes, some sequences which originate from cultures have an original morphologically linked ID, but if a phylogenetic tree shows it branching elsewhere, the taxonomic assignment of the sequence will be changed.

Response: Thank you for this comment. Yes, PR2 is not completely based on morphologically-identified specimens (nor is any existing sequence database). We have adjusted our description here by changing “morphologically-referenced sequence databases” to “curated sequence databases” (line 52).

Lines 53-54- I’m not so sure that morphology-based taxonomy is the gold standard in protistology. Most work in the last 10-20 years has focused on phylogenomics to denote new lineages of the eukaryotic tree of life and reclassify those that were taxonomically assigned morphologically in the past.

Response: We agree that phylogenomics is crucial for identifying new protist lineages and understanding their relationships, especially at high taxonomic levels. However, at the lower taxonomic levels that are the focus of this study (e.g., genus and species), there is not yet sufficient reference sequence data to accurately classify most taxa based on their sequence information alone. Reference databases currently lack sequence information for many formally-described protist species, and this situation is particularly true for the Radiolaria; thus, we feel that morphology does remain the “gold standard” for classifying lower-level taxa. As you point out, this standard is actively changing as new sequence data allow us to reclassify morphotaxa (and in fact, is somewhat the crux of our study), but for now we still must often rely on morphological descriptions for taxonomic work.

Line 224- This sentence is a little clunky. Please re-phrase.

Response: We have re-phrased the sentence, “Global scale genetic surveys of protistan species richness in the ocean have revealed surprisingly high abundance and diversity of radiolarian molecular taxa, exceeding more than an order of magnitude more taxa than might be anticipated from morphology based surveys of similar scale.” It now reads: “Global scale genetic surveys of marine protistan species richness have revealed such great diversity of radiolarian molecular taxa that they exceed the number of taxa in morphology-based surveys of similar scale by more than an order of magnitude.” (lines 244-247)

Methods:

Line 800- Did a single researcher perform the taxonomic classification? Were any other researchers consulted for the morphological classifications? What was the identification guide used?

Response: Yes, all taxonomic classifications were made by the same researcher (ST) who is formally trained in radiolarian taxonomy. Unfortunately, there is no comprehensive identification guide for extant polycystine Radiolaria in our study region, so several publications containing taxonomic descriptions and high-quality illustrations were used as reference guides. We have added the following text to clarify these points: “All taxonomic classifications were performed by ST based on formal descriptions and illustrations in several published literature sources, including: [11,24,38,49,55,57,60,62,65,67]. The curated online taxonomic resource Mikrotax (www.mikrotax.org [69]) was also consulted to confirm species identifications.” (lines 862-866)

Line 819- What parameters were used for quality and length filtering in DADA2? Which unique changes were made to the workflow to account for the long-reads?

Response: Thank you for catching this omission. We have added the following statements in lines 888-909: “Sequences were initially checked for primer matches and oriented in the forward direction using cutadapt, with the “linked adapter” setting (-g) and the parameters --revcomp --discard-untrimmed -e 0.1, to account for reverse-complemented reads, remove reads lacking both forward and reverse primer sequences, and limit the allowable error rate to 10%. PacBio Circular Consensus Sequence data must be corrected for orientation because reads can be output in either the forward or the reverse direction. DADA2 removePrimers function was then applied to double-check read orientation using the orient = TRUE parameter. The removePrimers function was also used to confirm that all reads contained primer matches (using the default maximum mismatch threshold of 2), remove any reads that did not contain primer matches, and then trim primer sequences from all passing reads. Next, the DADA2 filterAndTrim function was used to filter reads for quality and length. We used the parameters maxLen=1600 and minLen=1300, based on the expected length range of our target amplicons (1400-1500 bp). Following the filtering recommendations for long-read PacBio amplicon data (Callahan et al. 2019), we required that reads contain zero Ns, have a minimum quality score of 3, and a maximum expected error rate of 2 (maxN=0, minQ=3, maxEE=2). Within the DADA2 learnErrors function, we used the parameters “errorEstimationFunction=PacBioErrfun” and “BAND_SIZE=32,” as advised for PacBio long-read data (Callahan et al. 2019). To best detect rare sequence variants that may be shared by multiple specimens, the parameter pool=TRUE was used in executing the dada function. Finally, the removeBimeraDenovo function in DADA2 was used to identify and remove chimeric reads.”

I would suggest looking into the V4 region (most common region for metabarcoding studies) of these long read ASVs to see if they correspond to the same amount of variability, as this is what would really reveal if the different ASVs you see at the long-read level correspond to the variability at the short read level.

Response: Thank you for this suggestion. We have now added a comparative analysis of long-read sequence diversity and the extracted V4 regions from these sequences (see lines 135-145). There are new boxplot panels in Fig 1 (Fig 1e-h) to illustrate the richness of V4-only ASVs per specimen. We have also added an analysis of intra- and inter-species p-distance metrics, which were calculated using both long-read ASVs and extracted V4-regions only (see lines 415-456; Fig 7). New V4-region ASV occurrence charts have been added to S1 Dataset, and p-distance data is provided in the new S2 Dataset.

Results & Discussion:

Figure 1- a short title within or above each chart would be useful for data interpretation.

Response: We have added short titles above each chart in Fig 1.

Heatmaps- the light shading for high abundance and dark shading for low abundance seems reversed for me. I would suggest switching these around or changing the color scheme all together to be more distinct (might help to also remove the gray background?)

Response: We have removed the gray background and changed the heatmap color scheme so that light blue = less abundant and dark blue = more abundant.

Beautiful images and vizualization of the data in Fig 7, great job!

Response: Thank you!

Line 605- “not inconsistent” to “consistent”

Response: Change made (line 670).

Reviewer #2:

This manuscript entitled " Genetic variability in microbial eukaryotes reshapes marine biodiversity assessment in the age of amplicon sequencin" provided much information and discussion on the intragenomic and intraspecies genetic variation in one group of marine protists, the polycystine Radiolaria. The authors analyzed the combined microscopy and long-read 18S rRNA gene amplicon data sequencing data and made interesting and important findings that will have great implications for the interpretation of metabarcoding data. The manuscript is well-written, the text is well structured, clear and easy to understand. The results are well illustrated and the manuscript is accompanied by a large amount of supplementery material (photos of all 173 isolated specimens used in analyses, heatmaps illustrating ASV presence and abundance in all analyzed specimens, expanded phylogenetic tree based on the dominant ASVs and ect.). The authors have done extensive and important research. This manuscript is helpful to readers and researchers who are working on or beginning the metabarcoding study of marine protists.

Response: Thank you for your kind and supportive comments on the study and the manuscript.

My minor comments are addressed below:

Line 193. “…relative abundance (scaled from 0 to 1)…” If I understand correctly, relative abundance is calculated as n%/100%. Please clarify this in the caption to the figure. You use “%” in the text (lines 177-190). The discrepancy between the figure and the description is a bit confusing.

Response: We have now clarified this by changing the axis labels in Fig 2 from proportions to percentages, so that they match discussion in the text. The caption now also includes a brief definition of relative abundance.

Lines 207-209. … I would suggest to be more specific. I would suggest adding exact copy numbers of the 18S rRNA gene to the phrases... “astoundingly high…”, “…similar …” “…fewer copies…”

Response: This statement has been changed to reflect the range of copy numbers that have been reported in Collodaria cells by Biard et al. 2017 (lines 230-231).

Line 252. Table 1. The list is presented in alphabetical order by morphospecies name. I would suggest reformatting this list based on taxonomy (orders Nassellaria, Collodaria, Spumellaria). In my opinion, this way the data will look more structured and will correspond to its description.

Response: Table 1 has been reformatted according to order-level taxonomy.

It should also be noted that the main objective of this study is to investigate the intragenomic and intraspecies genetic variation in the polycystine Radiolaria and integrating morphological and molecular information. However, in my opinion it would be interesting to estimate the degree of similarity (or the degree of differences) between sequences of different ASVs of one sample/morphospecies and/or between different morphospecies (for example based on p-distans). I recommend this aspect needs to be addressed in the future. It will add useful information and clarity to the study. And it will probably allow us to determine the boundaries of intragenomic and intraspecies genetic variation among polycystine Radiolaria.

Response: Thank you for this suggestion. We agree that a quantitative difference metric would add an important dimension to our study. We have now computed p-distance metrics using MEGA 12 software, in order to examine ASV similarity within and between specimens and morphospecies. Results of this analysis are presented in the new Fig 7, and the full p-distance dataset is provided in S2 Dataset.

Reviewer #3:

The article is well written and structured, interpreting complex data with remarkable clarity. The organization of the content ensures a seamless flow, making the information accessible. It demonstrates the high intragenomic variance in polycystine Radiolaria and the importance of incorporating this variability in future metabarcoding survey.

Response: We thank the Reviewer for these supportive comments.

My main concern would be regarding the Table 3: More than half of input CCS reads was filtered out at the Quality & Length filtering step. However, about 89% reads passed after removing primers and filtering in the literature cited (Line 823). Is there any possible cause to the low passing rate? Was the Q score of raw data tested in other quality control tools? It'd be better if an explanation of the low passing rate is included.

Response: It is presumed that the low passing rate at the quality and length filtering step is primarily due to length rather than quality. The sequencing facility reported that 100% of our barcoded reads had Phred quality scores ≥Q20. Thus, PCR artefacts and/or off-target amplification are more likely to be the culprits of our relatively low passing rate. The mean read length of our raw demultiplexed data was 1252bp; however, based on known genomic characteristics of our target taxa, we expected reads of our target taxa to be approximately 1400-1500bp long. We thus required that reads passing our length filter be between 1300 and 1600 bp long, which alone would have removed approximately half of our reads for being too short. These relatively short reads may have originated from artefacts generated during PCR, or from contaminating taxa that happened to have shorter regions of the 18S gene and were unexpectedly amplified by our primers. We can speculate that the literature cited (Callahan et al. 2019) may have had a higher passing rate at this processing step because they used a more relaxed length filter (1000-1600bp; Callahan et al. 2019) and/or had lower rates of PCR artefacts and off-target amplification. We have added this explanation in lines 917-928.

Besides it, I only have a few minor comments and suggestions:

Line 29 – 33: It is a very nice summary of results. But it would be nice to expand a little if possible.

Response: Thank you for this suggestion. The end of the abstract has been expanded slightly to read: “Furthermore, every morphospecies analyzed displayed a range of different genetic signatures. Intraspecies genetic variability was expressed as specimens having different assemblages of ASVs, different dominant ASVs, or having no ASVs in common with other specimens of the same morphospecies. By integrating morphological and molecular information, we begin to parse the genetic richness of Radiolaria in ocean environments, as well as illuminate relationships between taxa, and their poorly-known life stages. Our findings emphasize the need to account for protists’ taxon-specific sequence variability, particularly their intragenomic and intraspecies genetic variation, in interpreting metabarcoding diversity survey data.”(lines 28-37)

Line 66 – 70: I believe there are also other research using metabarcoding as the Tara Oceans expedition and generated similar data. I think you can compare or add a few more references to back it up.

Response: While we a

---

## [Decision Letter · Decision Letter 1]

Genetic variability in microbial eukaryotes reshapes marine biodiversity assessment in the age of amplicon sequencing

PONE-D-24-55509R1

Dear Dr. Trubovitz,

We’re pleased to inform you that your manuscript has been judged scientifically suitable for publication and will be formally accepted for publication once it meets all outstanding technical requirements.

Kind regards,

Claudia Isabella Pogoreutz

Academic Editor

PLOS ONE

Additional Editor Comments (optional):

Reviewers' comments:

Reviewer's Responses to Questions

**Comments to the Author**

1. If the authors have adequately addressed your comments raised in a previous round of review and you feel that this manuscript is now acceptable for publication, you may indicate that here to bypass the “Comments to the Author” section, enter your conflict of interest statement in the “Confidential to Editor” section, and submit your "Accept" recommendation.

Reviewer #1: All comments have been addressed

Reviewer #2: All comments have been addressed

2. Is the manuscript technically sound, and do the data support the conclusions?

Reviewer #1: Yes

Reviewer #2: Yes

3. Has the statistical analysis been performed appropriately and rigorously? 

Reviewer #1: Yes

Reviewer #2: Yes

4. Have the authors made all data underlying the findings in their manuscript fully available?

Reviewer #1: Yes

Reviewer #2: Yes

5. Is the manuscript presented in an intelligible fashion and written in standard English?

Reviewer #1: Yes

Reviewer #2: Yes

6. Review Comments to the Author

Reviewer #1: (No Response)

Reviewer #2: The authors have satisfactorily addressed all my comments, revised the manuscript accordingly and added important clarifications.

Also, the authors conducted an additional analysis in the part of comparing the variability of the short V4 region (the most common region for metabarcoding studies) and the long read ASVs of 18S rRNA. This allowed them to draw important conclusions for interpreting metabarcoding data within marine protists.

The manuscript is much improved and can be accepted for publication.

7. PLOS authors have the option to publish the peer review history of their article (what does this mean? ). If published, this will include your full peer review and any attached files.

**Do you want your identity to be public for this peer review?** For information about this choice, including consent withdrawal, please see our Privacy Policy .

Reviewer #1: No

Reviewer #2: No

---

## [Editor Report · Acceptance letter]

PONE-D-24-55509R1

PLOS ONE

Dear Dr. Trubovitz,

I'm pleased to inform you that your manuscript has been deemed suitable for publication in PLOS ONE. Congratulations! Your manuscript is now being handed over to our production team.

Kind regards,

on behalf of

Prof. Claudia Isabella Pogoreutz

Academic Editor

PLOS ONE